# A Meta-Analysis of fMRI Studies of Youth Cannabis Use: Alterations in Executive Control, Social Cognition/Emotion Processing, and Reward Processing in Cannabis Using Youth

**DOI:** 10.3390/brainsci12101281

**Published:** 2022-09-23

**Authors:** Christopher J. Hammond, Aliyah Allick, Grace Park, Bushra Rizwan, Kwon Kim, Rachael Lebo, Julie Nanavati, Muhammad A. Parvaz, Iliyan Ivanov

**Affiliations:** 1Department of Psychiatry, Division of Child & Adolescent Psychiatry, Johns Hopkins University School of Medicine, Baltimore, MD 21287, USA; 2Department of Pediatrics, Division of Adolescent/Young Adult Medicine, Johns Hopkins University School of Medicine, Baltimore, MD 21287, USA; 3Wegner Health Sciences Library, University of South Dakota, Sioux Falls, SD 57105, USA; 4Welch Medical Library, Johns Hopkins University School of Medicine, Baltimore, MD 21287, USA; 5Department of Psychiatry, Icahn School of Medicine at Mount Sinai, New York, NY 10029, USA; 6Department of Neuroscience, Icahn School of Medicine at Mount Sinai, New York, NY 10029, USA

**Keywords:** adolescence, cannabis use, executive control, emotion processing, reward processing, brain activation, meta-analysis, fMRI, sex differences, abstinence

## Abstract

**Background**: Adolescent cannabis use (CU) is associated with adverse health outcomes and may be increasing in response to changing cannabis laws. Recent imaging studies have identified differences in brain activity between adult CU and controls that are more prominent in early onset users. Whether these differences are present in adolescent CU and relate to age/developmental stage, sex, or cannabis exposure is unknown. **Methods**: A systematic review and subsequent effect-size seed-based d mapping (SDM) meta-analysis were conducted to examine differences in blood-oxygen-level-dependent (BOLD) response during fMRI studies between CU and non-using typically developing (TD) youth. Supplemental analyses investigated differences in BOLD signal in CU and TD youth as a function of sex, psychiatric comorbidity, and the dose and severity of cannabis exposure. **Results**: From 1371 citations, 45 fMRI studies were identified for inclusion in the SDM meta-analysis. These studies compared BOLD response contrasts in 1216 CU and 1486 non-using TD participants. In primary meta-analyses stratified by cognitive paradigms, CU (compared to TD) youth showed greater activation in the rostral medial prefrontal cortex (rmPFC) and decreased activation in the dorsal mPFC (dmPFC) and dorsal anterior cingulate cortex (dACC) during executive control and social cognition/emotion processing, respectively. In meta-regression analyses and subgroup meta-analyses, sex, cannabis use disorder (CUD) severity, and psychiatric comorbidity were correlated with brain activation differences between CU and TD youth in mPFC and insular cortical regions. Activation differences in the caudate, thalamus, insula, dmPFC/dACC, and precentral and postcentral gyri varied as a function of the length of abstinence. **Conclusions**: Using an SDM meta-analytic approach, this report identified differences in neuronal response between CU and TD youth during executive control, emotion processing, and reward processing in cortical and subcortical brain regions that varied as a function of sex, CUD severity, psychiatric comorbidity, and length of abstinence. Whether aberrant brain function in CU youth is attributable to common predispositional factors, cannabis-induced neuroadaptive changes, or both warrants further investigation.

## 1. Introduction

Cannabis is the most commonly used illicit psychoactive drug by adolescents and consequently is the main drug that youth residing in the United States (U.S.) seek substance use treatment for [1]. Cannabis use among U.S. adolescents increases with age, such that in 2018 6.5% of 12–17 year olds and 22.5% of 18–21 year old reported current cannabis use, and 14% of 8th graders, 33% of 10th graders, and, 44% of all 12th graders reported lifetime cannabis use [2,3]. In addition, daily use of cannabis continues to rise for non-college youth in the U.S., reaching a record high of 13% [3]. As there are multiple reasons for these trends, recent changes in state policies such as decriminalization of cannabis and new forms of cannabis ingestion have also contributed to societal perception of cannabis as being less harmful, in turn contributing to the trends of increased availability and use by underage youth populations in the U.S [4].

The clinical relevance of increased cannabis availability and use among youth is related to the purported effects of such use on cognitive and neurobehavioral development [1]. There exists well documented evidence that cannabinoids precipitate short-term and long-term cognitive impairment, both in adults and adolescents. Acute cannabis intoxication, for example, is associated with transient mood perturbations such as euphoria, anxiety, and paranoia. Recent literature also suggests that adolescent cannabis users show signs of cognitive impairment in attention, executive functioning, memory, visual processing, and processing speed, as well as a decreased general and verbal IQ [5,6,7]. However, the effects of age and sex on the emergence and the clinical presentation of these deficits remain understudied. Further, existing evidence shows higher rates of substance use disorders (SUD) in adolescents with psychiatric disorders compared to peers without psychiatric disorders [8] and conversely higher rates of psychiatric comorbidities in adolescents with cannabis use disorder (CUD) compared to non-using youth [9]. In turn, longitudinal studies have shown that heavier and more persistent adolescent cannabis use is associated with an increased prevalence of psychiatric disorders in a dose-dependent manner [10]. For example, some have documented strong associations between adolescent cannabis use and an increased risk of developing psychotic symptoms and advancing towards a psychotic disorder [11,12,13]. In fact, daily and high potency cannabis use before the age of 15 poses a five-to-six-fold greater chance of developing a psychotic disorder compared to adolescent non-users [14]. Moreover, cannabis use is often concomitant with ADHD, mood and anxiety disorders in teens [15]. As the relationship between cannabis use and psychiatric comorbidities is bidirectional, one very relevant question pertains to the possibility that psychiatric disorders in childhood may represent predisposing factors for the development of early experimentation and problem use during adolescence.

In recent years, there has been a notable increase in reports that examine brain physiology through functional neuroimaging in adolescent cannabis users, examining the relationships between clinical symptoms associated with cannabis use and abnormalities in brain functions implicated in cognitive domains such as decision making, executive control and emotional and motivational processes. For instance, task-based functional magnetic resonance imaging (fMRI) studies showed alterations across distributed brain regions during tasks involving cognitive/executive control, memory and learning, reward processing, cannabis cue-reactivity, and emotional processing [7,16]. In relation to top-down executive functions such as working memory and attention, fMRI studies in adolescent cannabis users show altered brain activation in wide spread networks including various cortical regions [16,17,18,19]. Results, however, have been mixed. Whereas some studies show decreased activation in frontal, temporal, and parahippocampal brain regions, and increased activation in the parietal and medial prefrontal cortices and the bilateral insula, others show different activation patterns [7,17,18,20,21,22]. Similarly, brain networks, such as striatal reward circuit, dorsal lateral prefrontal (dlPFC) and anterior cingulate (ACC) cortices, related to motivation and emotional processing have shown both increased and decreased responsiveness during reward and cannabis cue-reactivity [23,24,25,26,27]. Taken together, although findings are mixed, these studies consistently show functional impairments in various brain regions, from the frontal cortical engagement in executive processing to that of the limbic system in emotional regulation, with long term adolescent cannabis use. One recent review of fMRI studies in adolescent cannabis users reported results suggesting that cannabis users may require the recruitment of more neural resources than non-using peers to achieve compatible performance on tasks across domains [28]. The authors stipulate that the frontal and parietal lobes were often identified as key regions engaged across tasks [17,18,19,24,27,29,30,31]. Another important observation relates to results from a few preliminary longitudinal studies indicating that there may be pre-existing activation differences, especially in prefrontal cortical (PFC) regions, in teens who will initiate cannabis use [28]. However, quantitative assessments (e.g., meta-analyses) of existing reports with respect to activation patterns during distinct cognitive processes (e.g., executive control vs. reward vs. emotion paradigms), severity of use and the contribution of developmentally sensitive factors such as age and childhood comorbid disorders and the implications of possible findings to clinical presentations are presently lacking.

Such quantitative assessments of regional activations that are either task-specific or consistently appear across tasks can be crucial in facilitating the early identification of individuals at risk for poor clinical outcomes and their associated health outcomes, and consequently in promoting the development of effective treatment and prevention strategies. Identifying neural targets of addiction treatment in youth can be crucial in refining existing treatments and informing the future development of diagnostic and prognostic biomarkers and brain circuit-targeted interventions [32]. Accordingly, the present study used a neuroimaging meta-analysis and meta-regression approach to investigate differences between cannabis using and non-using youth and investigate the relative influence of age, sex, cannabis dose and use severity, and psychiatric comorbidity on the pattern of brain activation during executive control, reward, and emotion processing tasks in adolescent-onset cannabis users. We qualitatively and quantitatively summarize functional neuroimaging studies that examine neural correlates of cannabis use in adolescents and young adults using an effect-size seed-based d mapping (SDM, also known as signed differential mapping) meta-analytic approach [33]. Coordinate based meta-analyses, including SDM, allow for the aggregation of neuroimaging data to reliably identify localization of anatomical and activation patterns that converge across studies. Predicting that the number of eligible studies would be small, limiting our ability to perform appropriately powered subgroup contrasts, we primarily sought to identify common brain circuits across and within executive control, reward and emotion processing domains that may be significantly correlated with cannabis use across different populations. Supplemental analyses were also conducted to ascertain the purported effects of age, sex, and cannabis dose and use severity, as well as the role of psychiatric comorbidity on neural activity in cannabis using youth.

## 2. Materials and Methods

A systematic review of peer-reviewed fMRI studies was conducted following the Preferred Reporting Items for Systematic Reviews and Meta-analyses (PRISMA) guidelines and methods [34]. A subset of studies from the review that included coordinate-level data was used in the SDM meta-analyses.

### 2.1. Search Strategy

We searched for studies indexed in the online databases PubMed/Medline, Cochrane, Embase, and Web Science from January 1990 to November 2019 using the following search terms: “Adolescent”[Mesh] OR “adolescent” OR “young adult” OR “youth” OR “teenager” AND “Neuroimaging”[Mesh] OR “Magnetic Resonance Imaging”[Mesh] OR “MRI” OR “functional MRI” OR “fMRI” OR “blood-oxygen-level-dependent” OR “BOLD” OR “brain activation” OR “brain activity” OR “brain function” OR “brain circuit” OR “neural” AND “Cannabis-Related Disorders”[Mesh] OR “cannabis use” OR “marijuana use” OR “cannabis abuse” OR “marijuana abuse” OR “cannabis dependenc*[tiab]” OR “marijuana dependenc*[tiab]” OR “cannabis addiction” OR “marijuana addiction” OR “cannabis use disorder” OR “marijuana use disorder” OR “cannabis*[tiab]” OR “marijuana*[tiab]” OR “marihuana*[tiab]” OR “Δ-9-tetra-hydrocannabidol” OR “THC”. Broad search terms were used to minimize the likelihood of the search not identifying all relevant studies. In addition, we manually scanned the references of included studies and cross-referenced relevant original research, reviews, and meta-analyses to identify studies that may have been missed by the search. Additional articles missed were also identified through subsequent search using parallel search terms but restricting the date window from December 2019 to January 2022.

### 2.2. Study Selection

Studies were selected if they met the following criteria: (1) included ≥ 10 participants; (2) participants were between the ages of 12 and 21 years; (3) used diagnostic criteria for cannabis use disorder (CUD) as specified by the DSM (DSM-IV or DSM-5) or described frequency or quantity of cannabis use (e.g., daily, weekly, etc.) in study participants; (4) used whole-brain fMRI and voxel-wise analyses; (5) reported within- or between-subject contrasts in BOLD signal across cannabis use (CU) and non-using typically developing (TD) control youth, or brain-behavior correlations between BOLD response and cannabis-related variables in a combined sample of CU and TD youth; (6) reported coordinates from the above whole-brain analyses in standardized anatomic space (i.e., Talairach or Montreal Neurologic Institute (MNI) space) and (7) provided information about the inclusion/exclusion (I/E) criteria, clinical characteristics, and demographics of the study sample.

Articles that studied adolescent CU within the context of co-occurring psychiatric disorders were included if studies also included active controls that did not use cannabis. Studies with young adult samples were included if they the mean age of participants was <22 years.

### 2.3. Data Extraction

Articles were extracted, organized, and reviewed using Covidence software (Covidence.org). Initial independent title and abstract evaluations were done to identify potential articles of interest by two authors (A.A. and K.K.). Data extraction accuracy showed high correspondence/agreement (>80%) between reviewers. Abstract evaluation was followed by an independent full-text review of articles. Group discussion was used to resolve uncertainties about inclusion criteria and finalize the list of articles included in the qualitative review and SDM meta-analysis.

To facilitate exploration and interpretation of results, relevant design features and sample characteristics from each study were extracted and used to assess study quality and characterize the degree of heterogeneity across studies.

To create the final list of studies included in the meta-analysis, we took a three-step approach: Studies identified with the above search that reported coordinates of activation differences between CU and TD control groups from whole-brain analyses in Talairach or MNI space were identified and marked for inclusion in the SDM meta-analysis. For whole-brain fMRI studies that provided insufficient information on coordinates, corresponding authors were contacted via email to determine if unthresholded statistical maps or coordinates and additional study details could be provided. Additionally, we searched NeuroVault (neurovault.org) using select search terms (from above) to try to find unthresholded statistical maps from the relevant studies. These approaches yielded five additional studies, with three providing unthresholded statistical maps. These maps plus peak coordinates from published data and from author correspondence were used for the meta-analysis.

### 2.4. Data Analysis

SDM meta-analysis procedures: All meta-analyses were carried out using the anisotropic effect-size signed differential mapping permuting subject images (SDM-PSI) software, v.6.21 (http://www.sdmproject.com; accessed on 21 March 2021). SDM meta-analysis is a statistical technique for meta-analyzing neuroimaging data that recreates voxel-level maps of effect sizes and their variance based upon T-maps [33]. In contrast to other meta-analytic approaches, SDM enables original statistical parametric maps and peak coordinates to be combined, and reconstructs positive and negative effects within the same statistical maps, preventing a voxel from appearing in opposite directions, and providing for a more accurate representation of the results.

Data coding and preparation for SDM meta-analysis: In preparation for the SDM meta-analysis, the following data coding steps were taken: For studies that met inclusion criteria, coordinates associated with CU groups or variables were manually recorded by three authors (A.A., C.J.H. and B.R.) and cross-checked for concordance. Coded anatomical foci were then double screened for accuracy. If the studies reported coordinates in either Talairach or MNI coordinates, a text file containing the reported coordinates and the t-score associated with those coordinates was created. If a study reported multiple experiments, the results were still reported in the same text file. *p*-values or z-values were converted into t-scores using the SDM Utilities calculator, otherwise the sign (i.e., direction) of their effect was reported as positive or negative. In addition, a table was made with the study identifier (main author), the t-score used to determine significance, and the sample size for the experimental and control groups. If a study reported a statistically significant corrected *p*-value, but provided insufficient information to transform the corrected *p*-value into a t-score, a t-score of 3.1 was used, giving a conservative estimate for this effect. Studies that had no significant peaks were also included. Each experimental study was categorized into one of eleven domains based upon the cognitive paradigm(s) administered during the fMRI scan session (see Appendix A). To prepare for meta-regressions and sensitivity tests, data on CU and TD youth’s age at time of scan, proportion of female participants, average days (current frequency) of past 30-day cannabis use (averaged over the past 3 months), duration of cannabis use (years), proportion of CU participants who met CUD criteria, proportion of CU participants who were tobacco smokers, proportion of CU participants with co-occurring alcohol use disorder (AUD), proportion of CU participants with psychiatric disorder that are frequently comorbid with CU (i.e., depression, anxiety disorders, ADHD, conduct disorder) were obtained for each study (see Appendix A). A number of studies included mean scores from the Cannabis Use Disorder Identification Test (CUDIT), a validated measure of CUD severity [35]. Mean CUDIT scores in CU participants from studies that reported it were also collected. These variables were qualitatively investigated related to study outcomes, with pre-selected variables used in a priori meta-regression analyses.

Meta-analysis procedures: The main analysis was conducted in four steps: (1) Meta-analysis across whole-brain fMRI studies: SDM meta-analyses were conducted on the statistical parametric maps showing group-level effects for each study to examine for unadjusted differences between youth with CU and matched TD youth. This was first done across all whole-brain fMRI studies (45 studies) to characterize whether “general” cannabis use effects could be observed between CU and TD youth across cognitive domains. (2) Cognitive Domain-specific Meta-analyses: Domain-specific subgroup meta-analyses were then conducted. This was done by stratifying each experimental study based upon the main cognitive domain probed and running separate SDM analyses using subgroupings of studies based upon these domains. Our main domain-specific analyses focused on executive control (16 studies), social cognition/emotion processing (9 studies), and reward processing (8 studies) domains. Exploratory analyses investigating other domain/subdomains were also conducted but were underpowered and should be interpreted cautiously. (3) Primary Meta-regression analyses: Next, linear meta-regressions were conducted focusing on age, sex, and cannabis use features, using a priori defined variables from each study as dependent variables to assess whether variation in these variables contributed to variance in the magnitude of BOLD signal differences observed between CU and TD youth across studies. Meta-regression analyses using mean age (years) and proportion of females from each study were conducted with the goal of determining whether BOLD signal differences in CU vs. TD youth were age-related and/or sex-dependent. As some studies have shown differences in health outcomes and brain-behavior relationships in CU as a function of cannabis dose, diagnostic status, and severity, we conducted meta-regressions focusing on cannabis-related variables including duration of cannabis use (years), the proportion of CU participants meeting CUD criteria, and mean CUDIT scores of CU participants. (4) Reliability and sensitivity tests: Lastly, we conducted a series of reliability and sensitivity tests described below. All models were thresholded using an uncorrected *p*-value < 0.005 consistent with other SDM meta-analyses [33].

Supplemental Subgroup Meta-analyses and Meta-regressions: Supplemental meta-analyses were conducted to examine the effects of length of abstinence on brain-behavior associations. This was done by stratifying studies into groups based upon the abstinence criteria used (ad-lib use to ≥12-h [10 studies], ≥24-h [11 studies], ≥48 to ≥72-h [7 studies], and ≥21-days or longer [15 studies] at the time of scan) and conducting subgroup meta-analyses. Sensitivity tests examining the impact of design-related factors on results were conducted focusing on different analytic approaches (e.g., studies using CU vs. TD group comparisons vs. studies examining brain-behavior associations with cannabis variables in combined samples), and population characteristics (e.g., studies that included and excluded youth with comorbid psychiatric disorders and studies restricting the sample to youth meeting CUD diagnostic criteria).

Reliability Analysis: A jackknife analysis was performed to establish reliability of our meta-analytic results. This was done for the primary SDM meta-analyses focused on all studies and on specific cognitive domains by removing a single experimental study and repeating the analysis in sequence.

Publication Bias Analysis: To tests for publication bias we created and interpreted funnel plots visualizing the effect-by-variance for the results of our primary meta-analysis, and used SDM’s Bias Test tool to quantitatively assess bias.

## 3. Results

### 3.1. Systematic Review and Qualitative Analysis

The search identified 1371 citations with 959 records excluded following title and abstract screen. Four hundred and twelve citations underwent full text review. Out of these citations, 45 fMRI studies examining whole-brain BOLD signal differences met all inclusion criteria and were included in the qualitative and quantitative analyses. Figure 1 is a PRISMA flow diagram depicting the search process. Results from the qualitative analysis are presented in Appendix A.

### 3.2. Study and Sample Characteristics

Forty-five eligible whole-brain fMRI studies were included in the SDM meta-analysis (see Table 1). This included 36 studies that involved a direct comparison of BOLD response contrasts between CU youth (n = 829; mean age [SD] = 18.94 [2.01] years; 30% female) and TD youth (n = 906; mean [SD] age = 19.06 [2.44] years; 35% female) and 9 studies that examined brain-behavior associations with CU variables from combined samples that included both CU and TD youth (n = 967; mean [SD] age = 16.24 [0.25] years; 38% female). In the meta-analytic sample, the mean ages (t = −0.23, *p* = 0.82) and proportion of participants who were of female sex (t = 1.02, *p*= 0.31) did not significantly differ between CU and TD groups. One study [36] examined BOLD differences between heavy CU and non-CU daily cigarette smokers and another study [37] compared cannabis users with and without major depression diagnoses. Three studies [18,31,38] sought to characterize dissociable effects from cannabis vs. alcohol on brain activity by comparing groups of cannabis only users, cannabis and alcohol co-users, alcohol only users, and healthy controls. Thirty-six of 45 studies included in the meta-analysis (80% of sample) controlled for alcohol use in their main analyses and 21 of 45 studies (47% of sample) controlled for tobacco use (Appendix A). Many studies excluded youth based upon psychiatric comorbidities or psychiatric medication use (Appendix A). The proportions of CU youth who were tobacco smokers, had co-occurring AUD diagnoses, and had comorbid psychiatric disorders (e.g., depression, anxiety, ADHD, conduct disorder) are presented in Appendix A.

### 3.3. Meta-Analysis: BOLD Signal Differences in CU vs. TD Youth within and across Domains

Primary meta-analytic results are summarized in Table 2. When conducted across all fMRI studies, at the conservative threshold of *p* < 0.005, the primary SDM meta-analysis identified no regions showing significant BOLD signal differences between youth with CU compared to TD youth. Rerunning the meta-analysis using a more lenient threshold of *p* < 0.025, resulted in the identification of a small cluster in the ventral medial (orbital) prefrontal cortex (vmPFC: 16 voxel cluster; MNI peak coordinate: x = 0, y = 42, z = −10; SDM Zmap = 2.11, *p* = 0.017) which showed greater activation in CU compared to TD youth. In meta-analyses stratified by cognitive domain, domain-specific differences between CU and TD youth were observed during executive control and social/emotional but not reward processing domains (Figure 2). During tasks of executive control, CU youth showed greater BOLD response in a small cluster localized to the right rostral medial prefrontal cortex (rmPFC: 5 voxel cluster; MNI peak coordinate: x = 4, y = 60, z = −4; SDM Zmap = 2.62, *p* = 0.004) when compared to TD youth. Given our a priori interest in executive control we further examined this result by rerunning the executive control meta-analysis at a more lenient threshold of *p* < 0.025 and conducting sensitivity tests. This meta-analysis identified the rmPFC cluster which was significantly larger (372 voxel cluster; MNI peak coordinate: x = 4, y = 60, z = −4; SDM Zmap = 2.62, *p* = 0.004) extending to the right ventral mPFC and to the left rostral and ventral mPFC and ACC. It also identified a second cluster centered in the right primary somatosensory cortex extending to the right supramarginal gyrus (SMG) and insula (239 voxel cluster; MNI peak coordinate: x = 44, y = −20, z = 14; SDM Zmap = 2.48, *p* = 0.007). Sensitivity and reliability tests on the main executive control meta-analysis and results from our executive control meta-regression analysis (see Section 3.4, Section 3.5 and Section 3.6) showed that the majority of the variance in this right rmPFC cluster was driven by CU vs. TD activation differences from executive control fMRI studies where all of the CU participants met DSM diagnostic criteria for CUD. During social cognition and emotion processing tasks, CU youth (compared to TD youth) showed decreased BOLD response in a midline cluster centered in the left dorsal medial prefrontal cortex (dmPFC) extending across the midline into the right dmPFC and posteriorly into the left and right dorsal ACC (dACC) (dmPFC/dACC: 64 voxel cluster; MNI peak coordinate: x = 2, y = 50, z = 22; SDM Zmap = −3.10, *p* = 0.00097). This dmPFC/dACC cluster remained significant across reliability tests, and when analyses were restricted to only the emotion processing domain. In the primary reward domain meta-analysis, no differences were observed between CU and TD youth in brain activation during reward processing. Restricting the reward domain meta-analysis to studies reporting reward feedback contrasts (6 studies) also showed no CU vs. TD group differences. Reliability tests of the main reward domain meta-analysis showed little variance in the null result but sensitivity tests related to psychiatric comorbidity and CUD diagnostic status both yielded significant effects (see Section 3.4 and Section 3.5). Exploratory subdomain meta-analyses showed no CU vs. TD group differences in brain activation during risky decision making or drug cue reactivity paradigms. Results from an exploratory working memory subdomain meta-analysis identified a small activation cluster in the right insula (Ins: 8 voxel cluster; MNI peak coordinate: x = 34, y = −20, z = 12; SDM Zmap = 2.70, *p* = 0.0035) for which CU youth showed greater brain activity compared to TD controls.

### 3.4. Meta-Regression Analysis: Age-Related, Sex-Related, and Cannabis-Related BOLD Effects

Age-related and Sex-related BOLD Effects: Results from the SDM meta-regression analyses across all studies examining the effect of mean age at the time of scan and proportion of female participants on BOLD differences between CU and TD youth did not yield statistically significant results. In meta-regressions stratified by cognitive domain, domain-specific effects of sex but not age were observed. Regarding age effects, domain-specific meta-regressions focusing on mean age of participants for executive control, social cognition/emotion processing, and reward domains each produced null findings. Regarding sex effects, domain-specific meta-regressions showed activation differences related to sex distribution during executive control but not social cognition/emotion processing or reward domains (Appendix A). Specifically, during executive control tasks, an increasing proportion of female participants was associated with a relative decrease in BOLD response in CU, relative to TD youth, in a left insula cluster (Ins: 19 voxel cluster; MNI peak coordinate: x = −38, y = 14, z = 2; SDM Zmap = −2.90, *p* = 0.002).

Cannabis-related BOLD Effects: Results from cannabis-related meta-regressions showed significant effects from variables indexing cannabis problem severity but not dose (Figure 3). In meta-regression analyses focused on CUD diagnoses, an increasing proportion of CUD diagnoses among CU participants was associated with a relative increase in BOLD response in CU vs. TD youth in the right rmPFC extending to the vmPFC during executive control tasks (rmPF: 11 voxel cluster; MNI peak coordinate: x = 8, y = 48, z = −8; SDM Zmap = 2.81, *p* = 0.002) and a relative decrease in BOLD response in CU vs. TD youth in the right insula during reward processing tasks (Ins: 38 voxel cluster; MNI peak coordinate: x = 36, y = 14, z = 0; SDM Zmap = −3.214, *p* = 0.0006). In meta-regressions focused on CUD severity, increasing CUDIT scores in CU participants were associated with a relative increase in BOLD response in CU vs. TD youth in the right insula during social cognition and emotion processing tasks (Ins: 44 voxel cluster; MNI peak coordinate: x = 40, y = 14, z = −12; SDM Zmap = 2.93, *p* = 0.0017). None of the other assessed cannabis-related variables were significantly associated with BOLD signal differences between CU and TD youth in meta-regression analyses.

### 3.5. Supplemental Subgroup Meta-Analyses and Meta-Regression Analyses

Abstinence-focused subgroup meta-analyses examined BOLD effects related to the length of abstinence at the time of scan assessed across tasks (see Appendix A and Appendix A). Results of these subgroup meta-analyses showed similar patterns of activation differences at 24-h and 48-h to 72-h abstinence but different activation patterns for other abstinence subgroups with no overlapping activation foci across early, intermediate, and late time periods within the first 30-days of abstinence. CU youth with ad-lib use to 12-h abstinence had increased BOLD response compared to TD youth in the right caudate and right thalamus, whereas CU youth with 24-h and 48-h to 72-h abstinence both had increased BOLD response in the right insula and right inferior frontal gyrus (IFG) compared to TD youth. CU youth with a longer period of abstinence (≥21 days) showed a divergent pattern of brain activation from other abstinence groups showing decreased BOLD response relative to TD youth, localized in the right and left dmPFC/dACC and the right precentral and postcentral gyri.

Our sensitivity tests investigated variance in brain outcomes related to study design and two relevant sample characteristics—psychiatric comorbidity and CUD diagnostic status. Focusing on study design and specifically different analytic approaches: A subgroup meta-analysis of the thirty-five studies that applied a CU vs. TD group comparison approach showed that CU youth (compared to TD youth) had increased brain activity in a cluster localized to the left IFG and anterior insula (IFG/aIns: 60 voxel cluster; MNI peak coordinate: x = −42, y = 20, z = −2; SDM Zmap = 3.23, *p* = 0.0006). In contrast a subgroup meta-analysis focusing on the nine studies that examined brain-behavior associations between BOLD response and cannabis use in combined samples of CU and TD youth showed that CU youth (compared to TD youth) had decreased brain activity in a bilateral dmPFC cluster (dmPFC: 197 voxel cluster; MNI peak coordinate: x = −4, y = 50, z = 34; SDM Zmap = −3.67, *p* = 0.0001). To examine the effects of psychiatric comorbidity on brain outcomes, we conducted a subgroup meta-analysis across all paradigms/domains focusing on twenty-three studies that used standardized psychiatric interviews to assess for psychiatric conditions in participants and applied strict I/E criteria related to comorbidity, resulting in a subgroup of studies that had CU and TD samples with limited/very low levels of psychiatric comorbidity. The results of this subgroup meta-analysis paralleled our main meta-analytic results across paradigms/domains yielding no significant findings at the *p* < 0.005 threshold. Rerunning this low psychiatric comorbidity subgroup meta-analysis at a more lenient threshold of *p* < 0.025 resulted in the identification of one small cluster in the left orbitofrontal cortex which showed greater activation in CU compared to TD youth. Domain-specific low psychiatric comorbidity subgroup meta-analyses showed variable effects that differed across domains. Restricting the executive control meta-analysis to studies with low levels of psychiatric comorbidity (15 studies) negligibly altered the results, which continued to show a cluster in the right rmPFC. When the reward domain meta-analysis was restricted to studies with low levels of psychiatric comorbidity (4 studies), one significant cluster emerged that was localized to the right insula (Ins: 34 voxel cluster; MNI peak coordinate: x = 36, y = 16, z = −4; SDM Zmap = 3.19, *p* = 0.0007), and showed greater BOLD response in CU, relative to TD youth. When the social cognition/emotion processing meta-analysis was restricted to studies with low levels of psychiatric comorbidity (4 studies) no significant clusters were identified.

Based upon results from our meta-regression analyses showing significant BOLD differences related to the proportion of CUD, we chose to conduct our planned supplemental subgroup meta-analysis on the five fMRI studies where all CU participants met criteria for CUD, even though this analysis was underpowered. Results of this subgroup meta-analysis (Appendix A) showed that adolescents with CUD had increased BOLD response compared to TD youth in one large right rmPFC/vmPFC cluster that extended medially encompassing left and right rmPFC, vmPFC, and dACC regions (vmPFC/rmPFC/dACC: 258 voxel cluster; MNI peak coordinate: x = 2, y = 50, z = −4; SDM Zmap = 3.35, *p* = 0.0004) and a second smaller cluster in the left inferior parietal lobule (IPL: 21 voxel cluster; MNI peak coordinate: x = −34, y = −50, z = 52; SDM Zmap = 3.15, *p* = 0.0008). Results were unchanged when the meta-analysis was rerun using an expanded subgroup that also included studies with samples where >70% of CU participants met CUD criteria.

Focusing on fMRI studies that used executive control paradigms, three studies had samples in which all CU participants met DSM-IV/5 criteria for CUD. Twelve of the other thirteen executive control fMRI studies did not provide detailed information on CUD diagnostic status. Running executive control domain meta-analyses in these subgroups revealed significant activation differences between CU and TD youth in the CUD subgroup but not the other subgroup. The executive control meta-analysis restricted to the CUD subgroup (i.e., the 3 executive control fMRI studies with CUD samples) identified a large cluster centered in the right rmPFC extending into bilateral ventral and dorsal mPFC regions (rmPFC/vmPFC/dmPFC: 269 voxel cluster; MNI peak coordinate: x = 10, y = 54, z = −6; SDM Zmap = 3.60, *p* = 0.0002) that showed greater BOLD response in CU compared to TD youth. This activation cluster directly overlapped with the vmPFC cluster identified in the primary meta-analysis across studies and the rmPFC/vmPFC clusters identified in the main executive control domain meta-analysis and meta-regression analysis, respectively. The subgroup meta-analysis restricted to the twelve other executive control fMRI studies that did not provide diagnostic information on CUD yielded no significant clusters. Similar subgroup meta-analyses related to CUD diagnostic status for social cognition/emotion processing and reward domains were not conducted as zero and one social cognition/emotion processing and reward processing fMRI studies, respectively, had samples in which the majority of CU participants met CUD criteria.

### 3.6. Reliability Analysis and Publication Bias Analysis

Jackknife analysis for the primary meta-analysis across all studies was not conducted given the null results for this analysis. Jackknife analyses for cognitive domain-specific meta-analyses identified no additional clusters at the *p* < 0.005 threshold that were consistently preserved (i.e., in >50% of analyses) when studies were sequentially removed from each analysis. Regarding the reward domain meta-analysis, no significant clusters were identified with sequential removal of each study during the jackknife procedure. Jackknife analyses of the executive control and social cognition/emotion processing domain results were largely preserved through most study combinations (see Appendix A). Brain activation differences between CU and TD youth in the rmPFC during executive control paradigms were preserved in 9 out of 16 study combinations. Brain activation differences between CU and TD youth in the dmPFC/dACC during social and emotion processing paradigms were preserved in 7 out of 9 study combinations. Results of our publication bias analyses (see Appendix A) showed no visual or statistical signal for publication bias. Funnel plots for our two primary meta-analytic results showed symmetric distribution of studies suggesting low evidence for bias and results from Bias testing were not significant for these analyses.

## 4. Discussion

This comprehensive meta-analysis examined convergent fMRI findings from tasks indexing regional brain activations within the executive function, social cognition/emotion processing, and reward processing domains in youth with cannabis use. Such an effort allowed for identification of consistent brain activation patterns that were associated with elevated propensity and severity for cannabis use, as well as their relationships with age, sex, cannabis use features, and comorbid psychopathologies in adolescents. The main findings here pertain to identifying differences in brain activation between CU and TD participants in the medial regions of the PFC during social cognition/emotion processing and to a lesser extent during executive function/cognitive control but not during reward processing. A second set of findings identified sex differences in brain activation during executive control, and brain activation differences across cognitive domains in relation to cannabis problem severity and psychiatric comorbidity. Sex differences were seen in insula activation during executive control tasks and showed reduced activation in CU, relative to TD youth, as a function of an increasing proportion of females. Regarding cannabis problem severity, activation differences between CU and TD youth were consistently observed when analyses were restricted to samples in which the majority of CU participants met DSM-IV/5 criteria for CUD or when CUD severity was considered. These analyses showed that higher CUD severity was associated with increased BOLD response in CU (relative to TD) youth, in right rmPFC and vmPFC during executive control tasks and decreased and increased BOLD response in the right insula during reward and social cognition/emotion processing tasks, respectively. Psychiatric comorbidity contributed some variance to brain outcomes with these effects varying across domains, suggesting complicated cannabis use-by-psychiatric comorbidity relationships. Supplemental analyses identified brain activation differences between CU and TD youth in the right caudate, thalamus, insula, precentral and postcentral gyri, and bilateral mPFC regions that varied as a function of the length of abstinence and showed some anatomic overlap and some regional divergence with CUD severity results. Collectively, these findings indicate that differences in brain activity between cannabis using and non-using adolescents in medial prefrontal, insula, and other cortical and subcortical regions implicated in control, social cognition/emotion processing, and motivation are present and vary as a function of sex, CUD severity, psychiatric comorbidity, and length of abstinence. Further, our results provide preliminary evidence that brain activation alterations in adolescent cannabis users with greater cannabis severity or CUD diagnoses are more consistent, more broadly distributed, and may reflect multidomain impairments across executive control, social cognition/emotion processing, and reward processing domains in these youth. Implications of these findings are discussed below.

The primary meta-analytic results from our study yielded significant group differences in the activation of regions within the medial PFC (e.g., rostral, ventral and dorsal mPFC) across all tasks, as well as individually in executive control and emotional processing domains. Specifically, across all tasks, higher vmPFC activation were observed in CU, compared to TD. When each cognitive domain was examined separately, in CU compared to TD, higher activations were observed in the right rmPFC during executive control tasks and lower activations were observed in bilateral dmPFC and dACC during social cognition and emotion processing tasks.

Our executive control domain meta-analysis identified cannabis-related activation differences in mPFC regions, consistent with prior studies in both adolescent [7,17,18,19,38] and adult [6,22] cannabis users. Given comparable between-group behavioral performance on cognitive tasks in most of these studies, the increased rmPFC activation observed in CU compared to TD youth points to inefficient cognitive processing and suggests that CU participants may have to recruit additional PFC resources to perform comparably to their non-using TD counterparts, which has been posited before [71,72,73,74,75,76]. Our executive control results may be better understood by examining them within the context of scientific findings of studies investigating changes in brain activation across childhood and adolescence during inhibitory control in normally developing youth and youth with familial SUD (e.g., individuals at SUD risk). For example, one report by Somerville et al., 2011 focused on typically developing youth suggests that activation during inhibitory control tasks tends to diminish from childhood to adolescence purportedly due to neurobehavioral maturation and improved impulse control [77]. Additionally, longitudinal studies in children of parents with and without AUD show that while unaffected youth show decreases in brain activation during inhibitory control tasks from ages 7 to 13, youth with familial AUD risk exhibit increased activation during the same task [58]. Together, these studies suggest that vulnerable youth may need to recruit ancillary brain regions/systems and expend greater effort on inhibitory control over the course of their development to perform similarly to their typically developing counterparts. By extension, this suggests that some of the mPFC activation differences we identified between CU and TD adolescents in our analyses may reflect preexisting vulnerability and also could be interpreted as a compensatory strategy to overcome possible pre-existing deficits. It is important to note that our sensitivity test restricting the executive control meta-analysis to studies where participants had very low levels of psychiatric comorbidity did not change the main results in this domain. This suggests that the variance in activation differences between CU and TD youth cannot be fully explained by the presence of comorbid psychiatric conditions, and indicates that some variance in brain activations could be related to cannabis exposure or other unmeasured confounders. Thus, our rmPFC result may be better explained in relation to cannabis use behaviors or pre-existing addiction vulnerability that is unrelated to psychiatric symptom expression in CU youth. Relatedly, during jackknife analyses, the rmPFC result identified in our main executive control meta-analysis exhibited moderate study-to-study variability, and was most consistently observed when reliability tests included studies with CUD samples. This suggests that the rmPFC activation differences during executive control tasks could reflect executive dysfunction in adolescents with more severe pathophysiology, such as those with heavier cannabis use or greater CUD severity [1,6]. Results from our executive control sensitivity tests and meta-regression analyses related to CUD diagnostic-status provide additional support for this hypothesis, which warrants further investigation.

Another possible explanation for both the executive control rmPFC finding and our other findings is that chronic cannabis exposure results in maladaptive neuroplasticity in these brain regions or that constitutional abnormalities in vulnerable youth may be further exacerbated by chronic cannabis exposure during adolescence, a sensitive developmental period. Related to this hypothesis, the spatial distribution of activation findings from the present study appear to correspond with brain regions known to have increased cannabinoid receptor 1 (CB_1_R) expression and are implicated in endocannabinoid (eCB) signaling [78]. Given this, we posit that the alterations in brain activation seen in adolescent cannabis users are more likely to be found in cortical and subcortical brain regions enriched for cannabinoid receptors, and may occur as a result of homeostatic alterations in brain physiology following chronic cannabis exposure, reflecting eCB system dysregulation in these youth [79,80]. Incorporating genetic and epigenetic data collection and analysis into future neuroimaging studies of CUD populations may provide additional clues into how eCB signaling and related neural processes are altered with cannabis use. Whether brain activation differences during executive control and emotion processing in CU youth compared to their TD counterparts are attributable to common predispositional factors, cannabis-induced neuroadaptive changes, or both warrant further investigation and may inform the development of future brain-based interventions.

For tasks that focused on social cognition/emotion processing domains, we found that CU youth showed less engagement of dmPFC/dACC regions compared to TD youth. This finding was highly reliable across jackknife analyses, and was also observed in our emotion processing subdomain meta-analysis. In contrast, the dmPFC/dACC cluster was not observed in our low psychiatric comorbidity subgroup meta-analysis, the results of which yielded no significant clusters. This sensitivity test result should be interpreted cautiously as 3 of the 4 studies included in the analysis used social cognition paradigms, which contributed less variance to our main social/emotional domain meta-analytic result compared to emotion processing studies. Still, it suggests that some of the variances in BOLD response between CU and TD youth during social cognition may be related to psychiatric conditions that co-occur with cannabis use in adolescents. Decreased activation in the dorsal PFC in cannabis using youth during emotion processing may reflect either inherent or drug associated emotional deficits for this group of adolescents. Emotion processing involves brain circuitry and subjective emotional experience undergo significant neurodevelopmental changes during adolescence [81]. Prior studies have shown that in comparison to adults, adolescents experience more frequent high-intensity positive and negative emotions, greater emotional intensity, and greater instability [82] and that adolescents primarily respond to emotional cues with increased activation of the subcortical circuitry [83]. This heightened emotional reactivity during adolescence is considered adaptive in many situations related to social group cohesion, communication, and support, but may also contribute to adolescent-specific developmental vulnerability to emotional disorders [82,83]. During the transition to young adulthood, prefrontal inputs to subcortical nuclei (e.g., amygdala) increase, allowing for better regulation of emotional responses [81,82,83,84], resulting in an expected increase of activation in related cortical brain regions and greater cortical-subcortical coupling. Accordingly, our finding of decreased dmPFC/dACC function points to deficits in emotional processing in adolescent cannabis users [85], which may in turn signify impairments in navigating the social environment [86,87]. As the dorsal mPFC is believed to support healthy emotional awareness/recognition and the dACC is implicated in emotion regulation [88], abnormal functioning of these regions may reflect impaired emotion recognition and regulation in cannabis using youth. [85,86,87] Reverse directional explanations for these associations are also possible. For example, emotion processing deficits could alternatively predate cannabis exposure and predispose youth to early cannabis engagement and affect-motivated cannabis use as a maladaptive strategy to cope with the challenges of social interactions [82].

Somewhat surprisingly, we did not find differences in activation during reward processing in our primary reward domain meta-analysis. There are several possible explanations for this. First, as already suggested, cannabis use in adolescents may be predominantly related to social and emotional domains and may have little effect on the reward brain system in cannabis users that do not have cannabis dependence (conversely see [89]). It is also possible that differences among reward tasks account for differences in the activation patterns, which when combined either do not survive the statistical threshold, counteract or cancel each other. Furthermore, the lack of differences in regional activation may be accounted by individual neurodevelopmental differences that moderate the signal in some yet poorly understood fashion resulting in seemingly null differences in brain activation among participating groups. Related to this assertion, it is notable that while we did not identify reward alterations in our primary reward domain meta-analysis, both our subgroup meta-analysis restricted to studies with very low levels of psychiatric comorbidity and our meta-regression analysis focused on cannabis problem severity identified significant reward-related brain activation differences between CU and TD youth. Interestingly, both of these activation differences mapped onto the right insula and were in opposite directions, such that insula activation was higher in CU, relative to TD youth, when the meta-analytic sample was restricted to studies with limited psychiatric comorbidity, and lower in CU, relative to TD youth, when examined in relation to the proportion of CUD diagnoses among CU youth. This suggests cannabis use-by-psychiatric comorbidity interaction effects on BOLD signal are present, and further vary as a function of CUD status. The directional pattern of these relationships suggests that CU youth without comorbid/co-occurring psychiatric conditions may exhibit dysfunctional reward processing showing hyper-responsiveness in the insula during reward feedback. However, when psychiatric comorbidity is not taken into account (as in our main reward domain meta-analysis), activation differences related to reward processing between CU and TD youth are more variable, and any effects of cannabis use on BOLD signal, if present, may be obfuscated by differences in underlying brain function related to comorbid psychiatric conditions that predate cannabis exposure and/or differences in the neural response to cannabis exposure in youth with and without comorbidities. Furthermore, these results indicate that when CUD diagnostic status is considered, another pattern emerges, whereby youth with high CUD severity also exhibit dysfunctional reward processing but in the opposite direction, showing hypo-responsiveness in the insula during reward feedback. Taken together, these findings point to parametric variation in reward responsiveness among CU youth as a function of psychiatric comorbidity and CUD diagnostic status. As such, differences in sample characteristics may explain some of the variability in outcomes from fMRI studies in the literature. Further exploration of brain correlates of cannabis exposure in combined samples of cannabis using and non-using youth with and without comorbid psychiatric conditions are needed.

One of the main objectives of this study was to use meta-analysis and meta-regression approaches to investigate the influence of age, sex, psychiatric comorbidity, and cannabis use features on BOLD differences between CU and TD youth. Partially consistent with our hypotheses, our results showed distinct activation differences related to sex, CUD severity, and length of abstinence but no effect of age, and mixed findings with regard to the effects of psychiatric comorbidity on brain function. Null findings for age effects on BOLD response may be due to design and analysis differences and limited variation in mean age of study samples, with early adolescent cannabis users (ages 12–13 years) being underrepresented among studies.

Regarding our psychiatric comorbidity effects, null results on the main effect of psychiatric comorbidity were not surprising given that our main meta-analytic results across studies (collapsed across domains) were also not significant when conservative thresholding was applied. Notably, the meta-analysis across studies with CU and TD groups matched for low psychiatric comorbidity did show greater activation in CU compared to TD youth in an orbitofrontal cortical region commonly reported in CU studies [22,23], when thresholded at *p* < 0.025. Additionally, domain-specific effects were seen for executive control, social cognition, and reward domains when psychiatric comorbidity was controlled for in the analyses. Specifically, the executive control meta-analysis showed no change in the main rmPFC cluster, and the dmPFC/dACC cluster identified in the main social cognition/emotion processing meta-analysis was no longer present when these analyses were rerun restricting the meta-analytic sample to studies with very low psychiatric comorbidity. Additionally, a difference in right insula activation between CU and TD youth during reward feedback was identified that was not present in the main reward domain meta-analysis when this analysis was rerun in a subgroup of studies with very low psychiatric comorbidity. These findings suggest that the activation differences between CU and TD youth observed in our meta-analytic report cannot be fully explained by the presence of comorbid psychiatric conditions. Rather, they point to psychiatric comorbidity playing a contributory role to variability in BOLD signal in CU and TD youth, in combination with other factors such as cannabis exposure, sex, and unmeasured confounders. Taken together, our results provide early evidence for cannabis use-by-psychiatric comorbidity interaction effects on brain outcomes during executive control, social cognition, and reward processing in CU youth, and suggest that these interaction effects may manifest as different BOLD response patterns in CU youth with and without psychiatric comorbidities. It is important to note that our approach to measuring psychiatric comorbidity effects in this study was non-specific. Thus, the comorbidity results presented here may better reflect the influence of ‘general’ psychopathology as opposed to specific psychiatric conditions on BOLD differences between CU and TD youth. These associations warrant further investigation, and should be examined in relation to both ‘general’ psychopathology and specific psychiatric symptom clusters and disorders in future studies.

Regarding sex effects, we identified a left insula cluster that varied between CU and TD youth as a function of sex distribution showing decreased BOLD response in CU relative to TD youth in executive control fMRI studies that had a higher proportion of female participants. This suggests that sex may moderate the relationships between cannabis exposure, executive function, and brain activity during adolescence. Previous studies have shown sex differences in neuropsychological functioning in adolescent and young adult cannabis users [90]. Our results are consistent with prior studies in cannabis using adolescents and adults that have shown alterations in brain structure, activity, and connectivity in cortical and cerebellar regions rich in CB_1_R in women that correlate with cognitive impairments and, in some cases, are directionally opposite from those found in men [91,92,93,94,95,96,97,98]. Of note, none of these prior reports identified sex-by-cannabis group differences in the insula, although one [98] did show higher cerebral blood flow in the insular cortex of males compared to females in both cannabis using and non-using control groups. Thus, our meta-analytic report is the first to describe sex-specific effects of cannabis exposure in this region in humans. A number of possible factors could explain this sex-by-cannabis effect. Sex differences in the sensitivity to adolescent cannabis exposure could partially explain this result. There is growing evidence for sexual dimorphism of the eCB system [99], including sex differences in CB_1_R expression in the insular cortex, which has been shown across species. These eCB differences emerge during puberty, and may render cannabis using girls more vulnerable to adverse health outcomes related to cannabis exposure [99]. In support of this, recent animal studies have shown sex-specific effects of adolescent cannabis exposure on cortico-striatal-limbic circuit maturation and long-term THC-related cognitive and affective alterations extending into adulthood [100]. Our sex-related BOLD effects could also be secondary to phenotypic differences between cannabis using boys and girls. Women typically start using cannabis later than men, but progress more quickly from first use to dependence (a phenomenon known as the ‘telescoping’ effect) [101]. Additionally, women who use cannabis report greater subjective psychoactive effects, cannabis withdrawal severity, cannabis-related problems, and have higher rates of comorbid mood and anxiety disorders compared to cannabis using men [102]. These sex differences in natural history of cannabis onset and clinical presentation may indicate divergent neurobiological underpinnings in cannabis using men and women. The extent to which these behavioral differences map onto brain differences and predate or are exacerbated by cannabis exposure warrants further investigation. Lastly, these results might also be explained by general neurodevelopmental differences between boys and girls [103], which could be further exacerbated by cannabis exposure. Given the growing clinical and preclinical evidence for sex differences related to cannabis exposure, research in this area is sorely needed and may inform sex-specific interventions for women and men with CUD.

Regarding our cannabis problem severity effects, we identified rostral and ventral mPFC and right insula clusters that varied between CU and TD youth as a function of CUD severity and affected cognitive domain (e.g., executive control, emotion, and reward). During executive control, CU (compared to TD) showed increased activation of the right rmPFC and vmPFC as a function of increasing proportion of CUD diagnoses in the sample. This parallels our main findings and suggests that impaired over-engagement of the rmPFC/vmPFC is associated with greater CUD severity [104]. Activation in the right insula was also associated with CUD severity, with the direction of this effect being domain dependent. During social/emotional processing tasks, we observed an increase in BOLD response in the right insula in CU compared to TD youth that varied as a function of increasing mean CUDIT score, such that higher mean CUDIT scores in CU participants were associated with higher insula BOLD response in CU, relative to TD youth. However, during reward processing tasks, we observed a decrease in BOLD response in this region as a function of increasing proportion of CUD diagnoses in the sample, such that a higher proportion of CUD in CU participants was associated with lower insula BOLD response in CU, relative to TD youth. This suggests that greater CUD severity is associated with insula hyper-responsiveness during social interactions and emotion processing and insula hypo-responsiveness during reward processing and motivated decision making. Together these findings across domains support a cognitive control deficit, reward deficit, and stress surfeit model of CUD severity in adolescents [105], consistent with Ernst’s triadic model of addiction vulnerability [53]. Our results are consistent with previous studies in adults showing CU vs. TD differences in cognitive control and reward-related brain function that are more prominent in dependent cannabis users compared to non-dependent users [69,106].

In our abstinence subgroup meta-analyses, we found evidence for distinct patterns of activation differences across domains between CU and TD youth that varied as a function of length of abstinence. Notably, we observed what appeared to be a subcortical-to-cortical gradient in activation differences across abstinence periods with chronic CU (relative to TD) exhibiting a shift from predominantly subcortical circuitry alterations during ad lib use to predominantly cortical circuitry alterations during longer periods of abstinence (e.g., Ad-lib to 12-h: caudate and thalamus; 24-h and 48-h to 72-h: insula and IFG; and >21 days: dmPFC/dACC and precentral and postcentral gyri). These findings converge with previous studies showing state-level differences in cortical and subcortical brain function as a function of ‘sated’ versus ‘abstinent’ states and time since last use in adolescent and adult cannabis users [66,78,107]. Taken in the context of recent advances in dynamic functional connectivity research, these findings could reflect distinct brain states emerging at specific periods of abstinence. Inter-individual variability in the magnitude and of the length of time/occupancy spent in these brain states might have prognostic significance related to relapse risk during a cessation attempt, and represent neural treatment targets [108]. Regarding the spatial distribution of our results, brain regions showing abstinence subgroup effects generally did not overlap with each other, but did show some regional overlap (e.g., insula) with brain regions related to CUD severity. The extent to which these cross-sectional analyses can be compared with one another to improve our understanding of neuroadaptive changes following cannabis cessation is unclear. This represents an important area of future inquiry that requires serial MRI scans within cannabis using individuals over time.

The existing literature on medial PFC regions in adolescents suggests that these are regions that undergo protracted development from childhood to late adolescence, making adolescence a critical period of neuroplasticity [77,81,84]. Although attempts have been made to identify separate and more specific functions linked predominantly to the rostral vs. dorsal mPFC, it seems very plausible that these regions are functionally interconnected and often activate during similar cognitive processes including forward planning, executive control, self-reflection and social awareness [88,104]. Results from the current meta-analysis reveal that CU adolescents show differences in activation primarily in regions linked to executive control and emotion processing, cognitive functions that underpin more complex behaviors such as social functioning [88]. Our report of alterations in brain activation possibly link to adolescents’ behavior in social milieu but not in relation to reward processing, which suggests that cannabis use in adolescence is perpetuated and facilitated by factors predominantly related to youth’s social environment and interactions and to a lesser extent by factors related to the rewarding properties of the drug [85,86,87]. If true, this speculation is consistent with existing evidence suggesting that therapies targeting family and social dynamics show higher efficacy in treating CUD in adolescents [109,110] and may further inform strategies for designing a therapeutic intervention that focuses on building social skills to facilitate peer interactions and minimize substance use.

This meta-analytic report has a number of important limitations. As our main aim was to conduct meta-analyses and meta-regressions, our study was reliant on the study methodology, analytic techniques, and measurements used in each of the included fMRI studies. Many of these studies had small sample sizes and there was significant heterogeneity in design, behavioral measurements, and handling of psychiatric comorbidity and co-use of alcohol and tobacco which likely contributed variance to study outcomes. Consequently, many of our planned subgroup and meta-regression analyses may have been underpowered to detect subtle differences in brain activation with small effect sizes between cannabis users and non-users. Our attempts to address these issues by contacting authors and searching imaging repositories were limitedly successful but did yield an additional five studies that were included in our analysis. Changes in data management, reporting, and data sharing practices (especially of full datasets and unthresholded statistical maps) are needed to advance meta-analytic inquiry in the field of neuroimaging [111]. In parallel, a consensus core outcome set (COS) of standardized measures related to adolescent cannabis use that incorporate indices of frequency, quantity, duration, dose, addiction severity (e.g., [112]) would reduce heterogeneity across studies and facilitate meta-analytic comparison across different cannabis using populations. No studies included in this meta-analysis reported biochemically quantified delta-9-tetra-hydrocannabinol (THC) or cannabidiol (CBD) levels as an index of cannabis exposure in CU participants, limiting our ability to investigate this relevant emerging domain. This is particularly salient given preliminary data showing divergent and at times opposing effects of THC and CBD on brain activity and cognitive, emotional, and reward processing in adults [113,114]. As such, future studies should measure THC and CBD exposure from cannabis use and relate these exposures to brain changes. Another major limitation of this meta-analytic report is that the cross-sectional experimental design of most included studies prevents us from inferring causality of these results. That is, it is impossible to determine if the reported group differences in activation predate the onset of cannabis use (i.e., if these are constitutional) or if they are “caused” or exacerbated by the cannabis use. Notably, this is an inherent concern in most published studies to date. However, long term longitudinal studies, such as the ABCD study, are poised to circumvent this issue and allow independent assessments of predispositions of cannabis use separately from the effects of cannabis use. For ease of interpretation and to facilitate quantitative comparisons, we focused specifically on task-based fMRI studies investigating BOLD response differences from task contrasts. Still, given our limited understanding of the relationship between the fMRI BOLD signal and the corresponding neuronal processes, it remains challenging to interpret the directionality of the findings. We chose not to include studies that used resting-state fMRI or functional connectivity outcomes in our meta-analyses. While this decision could be viewed as a limitation, it was made after closely reviewing functional connectivity studies identified in our initial search and determining that there were too few studies and too much heterogeneity in analytic approaches across these studies (e.g., seed-based vs. independent component analysis) to allow for an appropriately powered quantitative analysis. As the number of functional connectivity studies in cannabis using adolescent samples expands this will be an important next step that can build on the findings presented here and improve our understanding of brain network alterations at rest and during cognitive processes. Despite these limitations, our study also has some relevant strengths that are worth noting. It is one of the first meta-analytic studies to characterize brain activation differences between cannabis using and non-using individuals in a developmental sample of adolescents and young adults, and relate these activation differences to CUD severity and length of abstinence. In doing so, it identifies key neural targets to guide future research and theory development related to the pathophysiology of adolescent cannabis use. An additional strength is in the use of SDM meta-analysis and meta-regression techniques which enabled us to quantitatively assess relationships between BOLD response and age, sex, cannabis features, and psychiatric comorbidity in the sample and conduct sensitivity tests to clarify the extent to which brain outcomes were related to study design and sample characteristics.

## 5. Conclusions

In our view, this is the most comprehensive quantitative meta-analysis in adolescents with cannabis use completed to date. We conducted a series of comparisons, with our main meta-analytic results showing alterations in the activation of medial PFC regions, suggesting alterations in social cognitions/emotion processing and executive control but not reward processing in adolescent cannabis users: in other words—not the rewarding properties of cannabis per se but rather deficits in navigating social environment might play predisposing and/or perpetuating roles in cannabis use in adolescents. These main findings suggest that interventions providing focal training in social skills and emotion regulation in adolescents may be useful for cannabis prevention, early intervention, and might target neural alterations underlying social processing deficits in cannabis using youth populations. Results from our meta-regression analyses showed differences in insula activation that were sex-dependent and alterations in brain activation of adolescent cannabis users in the rmPFC/vmPFC and insula and in the caudate, thalamus, insula, dmPFC/dACC, and precentral and postcentral gyri that varied as a function of CUD severity and length of abstinence, respectively. Lastly, our sensitivity tests showed evidence of complex cannabis use-by-psychiatric comorbidity interaction effects on brain outcomes across domains in CU youth. These findings help to provide a more coherent picture of brain activation differences within and across domains in cannabis using adolescents and how sex, psychiatric comorbidity, and trait-level and state-level cannabis use features influence the BOLD signal during task-based fMRI in this population. Notably, activation differences were more prominent and distributed across mPFC and other brain regions and cognitive domains in CUD samples. Our meta-regression results lend additional support to the triadic model of addiction vulnerability—specifically showing that greater CUD severity among adolescent cannabis users was associated with a pattern brain activity reflecting impaired “top-down” cognitive control and altered “bottom-up” signaling with decreased reward responsivity and increased emotional reactivity. These findings give a clear path forward for the field and emphasize the need for both large-scale prospective longitudinal studies with standardized substance use measures (such as the ABCD study) and smaller well-controlled studies with serial fMRI scans in specific subgroups of cannabis users (e.g., female CU and CU with depression) at different stages of abstinence and recovery to characterize different aspects of cannabis-brain function relationships across development.

## Figures and Tables

**Figure 1 brainsci-12-01281-f001:**
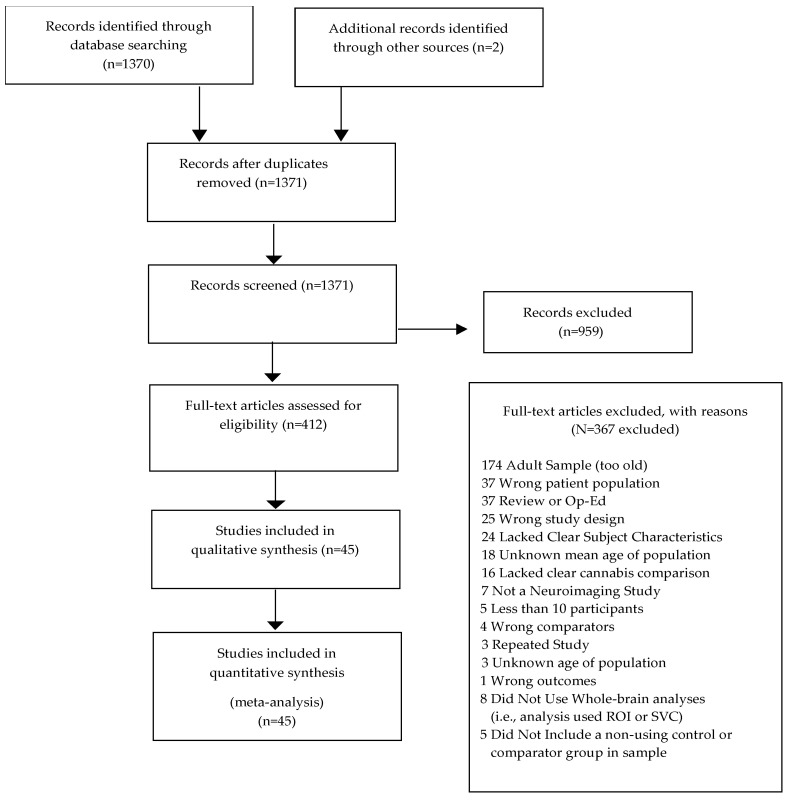
Flowchart outlining selection procedure of studies of fMRI differences.

**Figure 2 brainsci-12-01281-f002:**
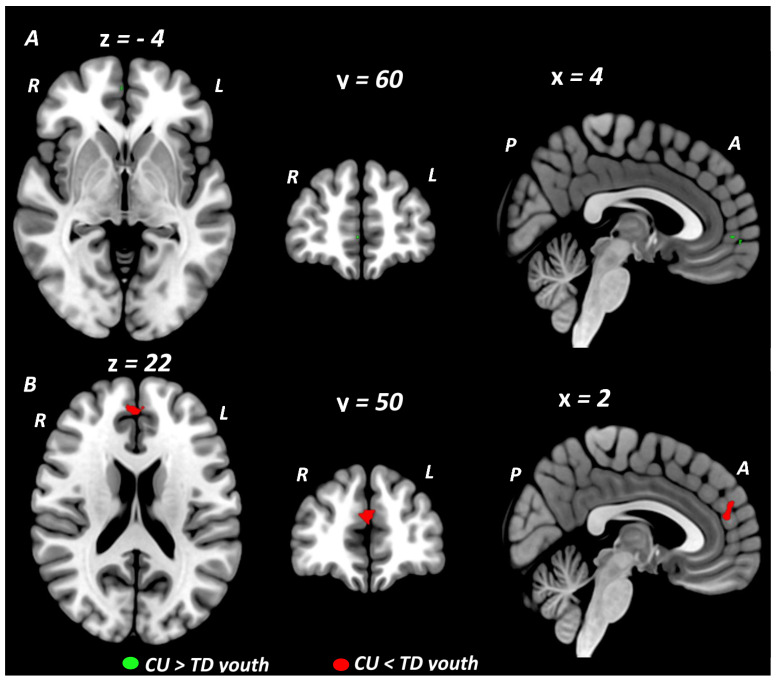
Meta-analysis Results Showing BOLD Signal Differences Between Cannabis Using and Typically Developing Youth during Executive Function/Cognitive Control (**A**) and Social Cognition/Emotion Processing (**B**). (**A**) Meta-analysis results for executive function/cognitive control studies. An increase in activation in CU youth compared to TD youth during executive control in the rostral medial prefrontal cortex (mPFC) (peak cluster of 5 voxels; MNI coordinates: x = 4, y = 60, z = −4) is shown in green. (**B**) Meta-analysis results for social cognition/emotion processing studies. A decrease in activation in CU versus TD youth during social cognition and emotion processing in the dorsal mPFC and dorsal anterior cingulate cortex (peak cluster of 64 voxels; MNI coordinates: x = 2, y = 50, z = 22) is shown in red. All results are thresholded at *p* < 0.005. Images visualized using MRIcroGL and presented on SDM template. Abbreviations: BOLD = blood-oxygen-level-dependent; CU = cannabis using; TD = typically developing; MNI = Montreal Neurologic Institute coordinates; mPFC = medial prefrontal cortex; R = right; L = left; A = anterior; P = posterior; CU = Cannabis Using; TD = typically developing.

**Figure 3 brainsci-12-01281-f003:**
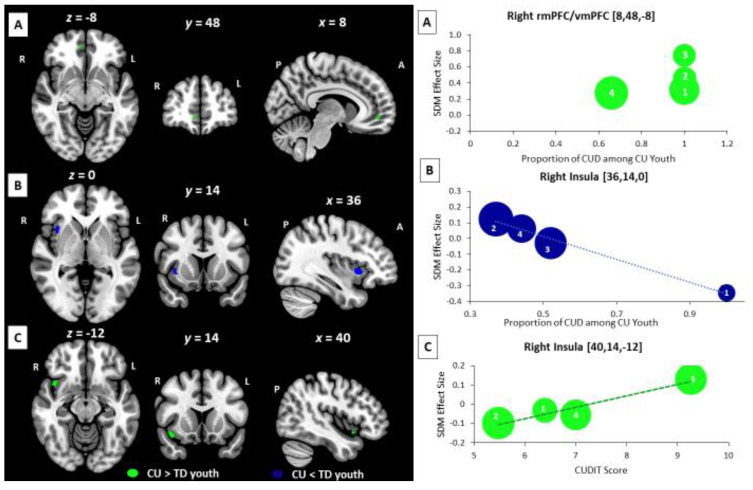
Meta-regression Results Showing an Association Between the Proportion of Cannabis Using Youth with Cannabis Use Disorder diagnoses and BOLD response during executive control (**A**) and reward processing (**B**) and an Association Between Mean CUDIT scores of Cannabis Using Youth and BOLD response during social cognition/emotion processing (**C**). Associations between the proportion of CUD diagnoses among CU youth and BOLD response differences in the rostral and ventral medial prefrontal cortex (CU > TD: rmPFC/vmPFC, 11 voxels, SDM-Z = 2.83) during executive control (**A**) are shown in green and in the right insula (CU < TD: Ins, 38 voxels, SDM-Z = −3.21) during reward processing (**B**) are shown in blue. Associations between mean CUDIT score among CU youth and BOLD response differences in right insula (CU > TD: Ins, 44 voxels, SDM-Z = 2.93) during social cognition and emotion processing (**C**) are shown in green. All results are thresholded at *p* < 0.005. Images visualized using MRIcroGL and presented on SDM template. Labeled meta-regression plots are presented to the right of each corresponding image (**A**–**C**). Effect sizes (SDM-estimates) used to create the meta-regression plots were extracted from the peak of maximum slope significance. The meta-regression SDM-estimate value is derived from the proportion of studies that reported BOLD signal changes near the voxel so it is expected that some values are at 0 or near +/− 1. Each included study is represented as a numbered dot, with the dot size reflecting relative total sample size of each specific study in comparison to the average total sample size of all studies included in the regression. For (**A**) the regression is plotted but the line is not shown as all but one study included in that analysis had a CUD proportion of 1.0. As this meta-regression result was driven by three studies (Schweinsberg et al., 2005, Behan et al., 2014, and Cyr et al., 2019) for which all CU participants met the criteria for CUD, this meta-regression analysis was rerun as a subgroup meta-analysis restricted to these 3 studies. The results of this analysis (detailed in Section 3.5) showed the same rmPFC/vmPFC cluster. Study key for 3A: 1 = Behan et al., 2014; 2 = Cyr et al., 2019; 3 = Schweinsburg et al., 2005; 4 = Schweinsburg et al., 2011. Study key for 3B: 1 = DeBellis et al., 2013; 2 = Aloi et al., 2019; 3 = Aloi et al., 2020; 4 = Aloi et al., 2021b. Study key for 3C: 1 = Blair et al., 2019; 2 = Leiker et al., 2019; 3 = Blair et al., 2021; 4 = Aloi et al., 2018. Abbreviations: A = anterior; P = posterior; L = left; R = right; rmPFC = rostral medial prefrontal cortex; CU = Cannabis Use; TD = Typically Developing; CUD = Cannabis Use Disorder; CUDIT = Cannabis Use Disorder Identification Test.

**Table 1 brainsci-12-01281-t001:** Summary of fMRI studies included in the Primary Meta-analysis.

Study	Sample Characteristics	Male Sex (%) by Group	Mean Age (Years) by Group	Quantity of CU among Participants	Sample Type	Abstinence at MRI Scan Session	MRI Scanner	Task Type	Task Contrast(s)	Analytic Method, MC, and Sampling Approach	Results of Whole-Brain Voxel-Wise Analysis:
Abdullaev et al., 2010 [39]	N = 14 Chronic CU youth and N = 14 matched healthy CON	CU: 71.4% CON: 71.4%	CU: 19.5 yrs. CON: 19.6 yrs.	CU group used on 132 days per year and for an average of 5 yrs.	Community	≥48-h	3.0 T	Attention Network task	Alerting effect: Center cue vs. No cue contrast; Orienting effect: Spatial cue vs. center cue contrast; Conflict effect: Incongruent vs. congruent contrast	Analysis: WB GLM FWHM: 6 mm MC: WB cluster corrected *p* < 0.05, Z > 2.3 Sampling: CU vs. TD group comparison	Alerting and Oriented effects: MJ > CON: None MJ < CON: None Conflict effec: MJ > CON: R lateral prefrontal cortex (BA 47) R supplemental motor cortex (BA 6) b/l lateral parietal corext (BA 40) MJ < CON: none
Acheson et al., 2015 [40]	N = 14 CU youth and N = 14 CON, ages 15–19 yrs	CU: 78.6% CON: 78.6%	CU: 17.3 yrs. CON: 17.6 yrs.	CU group used ≥5-days per week	Community	>12-h	3.0 T	Win/Loss Feedback task	Win vs. Neutral contrast; Loss vs. Neutral contrast	Analysis: WB and ROI, SEM FWHM: 5 mm MC: WB: k > 15 voxels, Cluster corrected *p* < 0.01, z ≥ 2.3 Sampling: CU vs. TD group comparison	Win vs. Neutral: MJ > CON: R middle frontal gyrus R caudate L middle frontal gyrus L caudate L claustrum R claustrum L middle frontal gyrus MJ < CON: None Loss vs. Neutral: MJ > CON: R middle frontal gyrus R posterior cingulate R anterior cingulate R claustrum L insula L claustrum L declive R declive MJ < CON: None
Aloi et al., 2018 [41]	N = 150 youth ages 14–18-years with variable levels of CUD and AUD severity recruited from residential program and community	Total Sample: 62% CU: 76% CON: 64%	Total Sample: 16.1 yrs. CU: 16.2 yrs. CON: 15.6 yrs.	Mean CUDIT score of Total sample = 7.0	Clinical & Community (combined sample)	≥30-days	3.0 T	Affective Stroop task	9 emotion-by-task contrasts were included based upon 3 emotional stimuli (positive vs. neutral vs. negative images) and 2 conditions (congruent vs. incongruent trials)	Analysis: WB + amygdala ROI, GLM + ANCOVA FWHM: 6 mm MC: WB: k > 19 voxels, *p* < 0.001 (via AFNI3dClustSIM) Sampling: Combined Sample	CUDIT-by-Task Condition effect was observed within the PCC, precuneus, IPL with Incongruent vs. Congruent contrast showing: Participants w/High-CUD symptoms > Participants w/Low/No-CUD symptoms: R PCC b/l precuneus R IPL R middle temporal gyrus L Culmen L cerebellum AUDIT-by-CUDIT-by-emotion-by-task condition interaction: Significant 4-way interaction was observed in the L IFG whereby AUDIT scores were negatively associated with IFG BOLD response to negative stimuli at low CUD levels (CUDIT < 4) but positively associated at high CUD levels (CUDIT > 27).
Aloi et al., 2019 [42]	N = 150 youth ages 14–18-years with variable levels of CUD and AUD severity recruited from residential program and community	Total Sample: 61%	Total Sample: 16.1 yrs.	Mean CUDIT score of Total sample = 7.31	Clinical & Community (combined sample)	≥30-days	3.0 T	Monetary Incentive Delay (MID) task	Reinforcement and Accuracy Effects: 4 reinforcement-by-accuracy contrasts were included based upon 2 reinforcement cues (reward vs. punishment cues) and 2 response outcomes (accurate vs. inaccurate response) on unmodulated BOLD response data.	Analysis: WB + ROI (striatum, ACC/dmPFC), GLM, ANCOVA FWHM: 6 mm MC: WB k > 26 voxels, voxelwise: *p* < 0.002, Cluster corrected *p* < 0.05 (via AFNI3dClustSIM) Sampling: Combined Sample	CUDIT-by-Accuracy Effect: Significant CUDIT-by-Accuracy effects were observed showing a strong negative correlation between CUDIT score and BOLD response in lingual gyrus and putamen during inaccurate compared to accurate trials. CUDIT-by-Reinforcement-by-Accuracy Interaction Effect: A significant CUDIT-by-Reinforcement-by-Accuracy interaction was observed within the R putamen and L ACC/dmPFC showing a negative correlation between CUDIT score and BOLD response during feedback on inaccurate punishment trials relative to all other outcomes. REW: In whole-brain voxel-wise analyses, no main effect of CUDIT was observed on BOLD response during reward feedback of accurate trials in the sample
Aloi et al., 2020 [43]	N = 104 youth ages 14–18-years with variable levels of CUD and AUD severity recruited from residential program and community	Total Sample: 64%	Total Sample: 16.1 yrs.		Clinical & Community (combined sample)	≥30-days	3.0 T	Passive avoidance task	4-stimulus-by-feedback contrasts were included based upon 2 stimulus types (high punishment probability stimulus vs. high reward probability stimulus) and 2 feedback outcomes (reward vs. punishment)	Analysis: WB, GLM, ANCOVA FWHM: 6 mm MC: WB k > 16 voxels, voxelwise: *p* < 0.001, cluster corrected *p* < 0.05 (via AFNI3dClustSIM) Sampling: Combined Sample	In whole-brain voxel-wise analyses no regions showed either a CUDIT-by-Feedback effect or an AUDIT-by-CUDIT-by-Feedback interaction effect that survived correction for multiple comparisons.
Aloi et al., 2021a [44]	N = 141 youth ages 14–18-years with variable levels of CUD and AUD severity recruited from residential program and community	Total Sample: 57%	Total Sample: 16.3 yrs.		Clinical & Community (combined sample)	≥30-days	3.0 T	Comparative optimism (CO) task	BOLD response was measured while participants were presented with future events that varied in valence and intensity and were asked to rate the probability of those events occurring. Contrasts of Interest: 4 valence-by-intensity contrasts based upon 2 valences of future events (positively valenced vs. negatively valenced) and 2 levels of intensity (high-intensity vs. low-intensity) future events.	Analysis: WB, GLM, ANCOVA FWHM: 6 mm MC: k > 23 voxels, voxelwise: *p* < 0.001, cluster corrected *p* < 0.05 (via AFNI3dClustSIM) Sampling: Combined Sample	CUDIT-by-Intensity effect: R/L subgenual ACC R/L PCC R superior temporal gyrus L fusiform L culmen R putamen In total sample, greater CUDIT scores were associated with greater differential BOLD responsiveness within the R/L sg ACC, R/L PCC, R superior temporal gyrus, L fusiform, L culmen, and R putamen to high-relative to low-intensity future events. CUDIT-by-Valence-by-Intensity effect: L precentral gyrus R/L cuneus L Occipital cortex R culmen In the total sample, there was a significant negative relationship between CUDIT scores and differential BOLD responsiveness within the precentral gyrus, cuneus, and occipital cortex to high-intensity relative to low-intensity negative future events. AUDIT-by-CUDIT-by-Intensity interaction effect: R/L rmFC
Aloi et al., 2021b [45]	N = 128 youth ages 14–18-years with variable levels of CUD and AUD severity recruited from residential program and community	Total Sample: 61%	Total Sample: 16.7 yrs.		Clinical & Community (combined sample)	≥30-days	3.0 T	Novelty task (three-armed bandit paradigm)	RPE during Explore vs. Non-Explore trials contrast Behavioral outcome: Novelty Propensity (NP) score Main analysis examined RPE-modulated BOLD responsiveness during explore vs. non-explore trials using a one-way ANCOVA with the following between-subject variables: AUDIT score, CUDIT score, NP score, Sex, AUDIT-by-NP interaction, and CUDIT-by-NP interaction.	Analysis: WB, GLM, ANCOVA FWHM: 6 mm MC: WB k > 17 voxels, voxelwise: *p* < 0.001, cluster corrected *p* < 0.05 (via AFNI3dClustSIM) Sampling: Combined Sample of CU and TD youth	Main effect of CUDIT: No main effect of CUDIT on RPE-modulated BOLD response was observed. CUDIT-by- NP score: There was a significant CUDIT-by-NP interaction within IPL and Cerebellum—whereby NP score was positively associated w/ RPE modulated BOLD response in youth with low CUDIT scores (<6) within IPL and cerebellum and was negatively associated with RPE modulated BOLD response in youth with high CUDIT scores (>12) [i.e., CUD < CON] in the cerebellum. CUDIT-by-NPS-by-Explore condition: Significant CUDIT-by-NP-by-explore interaction within dmPFC, IPL, and STG whereby NP score was positively associated w/RPE modulated BOLD response in youth with low CUDIT scores (<6) within dmPFC, STG, and IPL, and was negatively associated with RPE modulated BOLD response in youth with high CUDIT scores (>12) [i.e., CUD < CON] in the dmPFC, STG, and IPL.
Ames et al., 2013 [46]	N = 13 heavy MJ users and N = 15 healthy CON young adults (ages 19–25-years)	MJ: 85% CON: 33%	MJ: 21.2 yrs. CON: 20.3 yrs.	MJ group had >300 use episode over prior 3-years	Community	≥24-h	3.0 T	Marijuana Implicit Association task	Compatible association vs. fixation contrast; Incompatible association vs. fixation contrast	Analysis: WB + ROI, GLM FWHM: 4 mm MC: WB k > 30 voxels, voxelwise: *p* < 0.005, cluster corrected *p* < 0.05 w/MCS Sampling: CU vs. TD group comparison	Compatible associations: MJ > CON: L caudate R caudate R putamen L putamen R inferior frontal cortex MJ < CON: none Incompatible associations: MJ > CON: none MJ < CON: R inferior frontal cortex
Behan et al., 2014 [47]	N = 17 heavy CU youth and N = 18 non-using CON between ages 15–18-years	CU: 94.1% CON: 94.4%	CU: 16.5 yrs. CON: 16.1 yrs.	Cannabis user smoked 178.4 joints on average and smoked 4168 joints in their lifetime.	Clinical (drug treatment center in Dublin, IR)	≥12-h	3.0 T	Go/No-Go task	Successful inhibition of prepotent response (STOP) trials and Unsuccessful inhibition of prepotent response (ERROR) trials	Analysis: WB: FWHM: 4.2 mm MC: WB: Voxel-wise: *p* < 0.005, t = 3.01; Cluster-level: k > 277 μL, Cluster corrected *p* < 0.05 based on MCS Sampling: CU vs. TD group comparison	MJ vs. CON: No group differences
Berk et al., 2015 [48]	N = 15 adolescents (ages 15–17-yrs. With SUD related to cannabis and/or alcohol and N = 18 matched CON	SUD: 67% CON: 61%	SUD: 16.6 yrs. CON: 16.5 yrs.	73% of SUD group met criteria for CUD	Community (California high schools)	≥72-h		Aversive inspiratory breathing load task	Anticipation vs. baseline contrast; breathing load vs. baseline contrast	Analysis: WB + insula and ACC ROI FWHM: 4 mm MC: WB k > 768 µL (12 voxesl), cluster corrected *p* < 0.05 (via AFNI3dClustSIM) Sampling: CU vs. TD group comparison	Main group effects across phases: SUD > CON: None SUD < CON: L precentral gyrus L superior temporal gyrus Group effects-by-phase: Anticipation Phase: SUD > CON: None SUD < CON: Posterior insula Parahippocampal gyrus Superior temporal gyrus Breathing load Phase: SUD > CON: Posterior insula Middle frontal gyrus Uncus Middle temporal gyrus Anterior insula Inferior frontal gyrus SUD < CON: None
Blair et al., 2019 [49]	N = 87 youth ages 14–18-years with variable levels of CUD and AUD severity recruited from residential program and community	Total Sample: 49%	Total Sample: 16.48 yrs.	Mean CUDIT score of Total Sample = 6.4	Community/Clinical	≥30-days	3.0 T	Looming threat task	Direction (Looming vs. receding) by Type (animal vs. human) by Emotion (threatening vs. neutral) contrasts; main contrast: Looming vs. receding threat contrast; secondary contrast: threatening vs. neutral stimuli contrast	Analysis: WB, GLM, ANCOVA FWHM: 6 mm MC: WB k > 23 voxels, voxelwise: *p* < 0.001, cluster corrected *p* < 0.05 (via AFNI3dClustSIM) Sampling: Combined Sample of CU and TD youth	CUDIT-by-Direction contrast: Increasing CUD symptoms were associated with reducing BOLD response differentiation of looming vs. receding stimuli in rostromedial frontal cortex (rmPFC), L fusiform gyrus, cerebellum Traditional group-based analysis: Looming vs. receding threat contrast: MJ < CON: Rostromedial PFC Threatening vs. Neutral contrast: No main or interaction effect related to CUD symptoms.
Blair et al., 2021 [50]	N = 102 youth ages 14–18-years with variable levels of CUD and AUD severity recruited from residential program and community	Total Sample: 66%	Total Sample: 16.5 yrs.	Mean CUDIT score of Total Sample = 9.3		≥30-days		Retaliation task (variation on ultimatum game)	Primary contrast of interest: BOLD response when retaliating to unfair offers Task included 3 phases (offer vs. decision vs. outcome phases), 4 offers with variable levels of fairness or unfairness (fair vs. 3-levels of unfair offers), and 4 decision response options (accept offer vs. reject offer and punish partner by spending $1, $2, or $3 as punishment dollars).	Analysis: WB, GLM, ANCOVA FWHM: 6 mm MC: WB k > 19 voxels, voxelwise: *p* < 0.001, cluster corrected *p* < 0.05 (via AFNI3dClustSIM) Sampling: Combined Sample of CU and TD youth	In whole-brain voxel-wise analyses no regions showed either a CUDIT-by-phase effect or an AUDIT-by-CUDIT-by-Phase interaction effect that survived correction for multiple comparisons.
Claus et al., 2018 [31]	N= 39 MJ users, N = 90 MJ + ALC users, N = 23 ALC users, and N = 37 healthy CON adolescents ages 14–18-yrs.	MJ: 72% MJ + ALC: 88% ALC: 61% CON: 54% 129 (82.3%)	MJ: 16.0 yrs. MJ + Alc: 16.3 yrs. ALC: 16.4 yrs. CON: 16.1 yrs.	MJ group used approximately 14.6 days in the past 30-days	Justice System (alternative to incarceration program ©n SW United States)	≥24-h	3.0 T	BART risky decision-making task	Mean risk vs. Mean non-risk contrast (mean level of response for risky vs. riskless decisions across balloons); Linear risk vs. Linear Non-risk contrast (difference in partial correlation coefficient between BOLD signal and # pumps during risky vs. riskless choices)	Analysis: WB, ANOVA FWHM: 5 mm MC: WB: voxel threshold Z > 2.33, corrected cluster *p* < 0.025 Sampling: CU vs. TD group comparison	Mean risk vs. non-risk contrast: MJ + AUD > CON: none MJ + AUD < CON: b/l ventral striatum, thalamus, brain stem L putamen, insula, IFG Linear risk vs. non-risk contrast: MJ + AUD > CON: none MJ + AUD < CON: L Pre/Post-central gyrus, SPL, R putamen, caudate, insula, dACC/SMA
Cousijn et al., 2012a [24]	N = 31 frequent CU, N = 20 sporadic CU, and N = 20 non-using CON youth	Frequent CU: 65% Sporadic CU: 65% CON: 64%	Frequent CU: 21.3 yrs. Sporadic CU: 22.1 yrs. CON: 22.1 yrs.	Frequent CU reported using cannabis > 10 days per month for past-2-years and not having CUD treatment. Sporadic CU had between 1 and 50 lifetime CU episode.	Community (Amsterdam)	≥24-h	3.0 T	Visual Cannabis Cue reactivity task	Cannabis vs. neutral cue contrast	Analysis: WB, GLM, Regressions FWHM: 5 mm MC: WB: Corrected cluster pFWE < 0.05, z > 2.3 Sampling: CU vs. TD group comparison	Cannabis vs. neutral cue contrast: WB analysis: No Frequent MJ or Sporadic MJ vs. CON group differences. ROI: analysis: Frequent MJ > Sporadic MJ and CON: VTA Sporadic MJ vs. CON: No group differences Dependent MJ users > Non-dependent MJ users: ROI analysis: b/l ACC b/l OFC L putamen b/l caudate b/l Nucleus accumbens WB analysis: L middle frontal gyrus L temporal pole
Cousijn et al., 2012b [25]	N = 33 heavy CU young adults and N = 36 matched healthy CON young adults ages 18–25 yrs. who completed an MRI session at baseline and then a follow-up assessment at 6-months.	CU: 64% CON: 64%	CU: 21.3 yrs. CON: 22.2 yrs.	Heavy CU used > 10 days per month over the past-2-yrs.	Community (Amsterdam)	≥24-h	3.0 T	Approach bias Stimulus Response compatibility (SRC) task	Task includes approach, avoid, and baseline blocks/trials and uses cannabis and neutral images. Primary contrast: cannabis approach-bias obtained by subtracting avoid block (avoid-cannabis & approach-control) from approach block (approach-cannabis & avoid-control). Additionally four secondary condition vs. baseline contrast(s) were also investigated.	Analysis: WB, GLM FWHM: 5 mm MC: WB: Cluster corrected *p* < 0.05, Z > 2.3 Sampling: CU vs. TD group comparison	Approach block > Avoid block: Group Comparisons: No significant MJ vs. CON group differences were observed in approach-bias BOLD response. Within MJ group association analyses: Lifetime cannabis use positive correlation: L parahippocampal gyrus R amygdala b/l Occipital cortex b/l Cerebellum R insula R inferior frontal gyrus b/l medial frontal gyrus R precuneus L supramarginal gyrus Change in CUDIT negative correlation: R dlPFC b/l ACC
Cousijn et al., 2013 [51]	N = 32 heavy CU youth and N = 41 matched non-using CON youth completed MRI scan at baseline and had a follow-up assessment at 6-months	CU: 66% CON: 63%	CU: 21.4 yrs. CON: 22.2 yrs.	CU group used cannabis an avg. of 4.0 days per week and had a mean CUDIT score of 12.2 at baseline.	Community (Amsterdam)	≥24-h	3.0 T	Iowa Gambling task	Decision making phase: disadvantageous vs. advantageous choices contrast Feedback phase: win vs. loss feedback	Analysis: WB, GLM FWHM: 5 mm MC: WB: Corrected cluster pFWE < 0.05, z > 2.3 Sampling: CU vs. TD group comparison	Decision making: Disadvantageous vs. advantageous contrast: No MJ vs. CON group differences in BOLD response at baseline visit Reward Feedback phase: Win > Loss feedback: MJ > CON: R orbitofrontal cortex R insula L posterior superior temporal gyrus MJ < CON: None
Cyr et al., 2019 [52]	N = 28 CU youth and N = 32 healthy CON youth ages 14–23-yrs.	CU: 61% CON: 53%	CU: 19.3 yrs. CON: 18.9 yrs.	CU group used > 2 times per week	Community/Clinical	≥12-h	3.0 T	Simon Spatial Incompatibility Task	Incongruent (I) vs. Congruent©) contrast	Analysis: WB, Multilevel Regressions FWHM: 8 mm MC: WB voxelwise: *p* < 0.001, cluster corrected pFWE < 0.05 in SPM Sampling: CU vs. TD group comparison	I vs. C contrast: MJ > CON: none MJ < CON: R orbitofrontal cortex (lateral) R inferior frontal gyrus (orbitalis) L thalamus B/l orbitofrontal cortex (medial) L anterior cingulate cortex R supramarginal gyrus R postcentral gyrus R Rolandic operculum
Debellis et al., 2013 [53]	N = 15 adolescents with CUD in post-treatment remission with >30-days abstinence compared to N = 18 healthy TD controls and N = 23 CON with psychiatric comorbidities, all groups ages 13–17-yrs.	CUD: 100% TD CON: 100% Psychiatric CON: 100%	CUD: 16.4 yrs. TD CON: 16.0 yrs. Psychiatric CON: 15.4 yrs.	All CUD youth received treatment, were in full remission, and had been >30 days abstinent at scan session	Community/Clinical (CUD and Psychiatric CON from clinic and TD CON from community)	≥30-days	3.0 T	Decision Reward Uncertainty task	DM: Decision-making phase: Uncertain reward risk vs. known reward probability risk and no-risk contrast; REW: Outcome phase: Reward vs. No-reward outcomes during risky decision trials (behavioral and reward risk trials)	Analysis: WB + ROI, GLM FWHM: 5 mm MC: Cluster corrected pFWE = 0.05 Sampling: CU vs. TD group comparison	Uncertain risk vs. known risk DM contrast: CUD > CON with psychopathology: L superior parietal lobule and left lateral occipital cortex, precuneus L superior parietal lobule L lateral occipital cortex L precuneus R precuneus; Reward Outcome Contrast: CUD > CON with psychopathology: None CUD < CON with psychopathology: L frontal lobe/middle frontal gyrus/OFC L frontal lobe/MFG L middle frontal gyrus L frontal pole/OFC L superior frontal gyrus L middle frontal gyrus
Ford et al., 2014 [37]	N = 15 MJ using youth, N = 14 MDD + MJ use youth, N = 15 MDD youth, and N = 17 healthy CON youth ages 16–25-yrs.	MJ: 67% MDD + MJ: 71% MDD: 13% CON: 35%	MJ: 20.2 yrs. MDD+ MJ: 19.9 yrs. MDD: 19.7 yrs. CON: 20.0 yrs.	MJ group and MDD + MJ groups used on 22 and 21 days in the past month respectively	Community/Clinical		3.0 T	Passive music listening task	Preferred music selection vs. neutral music contrast	Analysis: WB, GLM, ANCOVA, regressions FWHM: 8 mm MC: pFDR < 0.05 Sampling: CU vs. TD group comparison	Preferred vs. neutral contrast: MJ > Other Groups: None MJ < Other Groups: None Preferred vs. neutral contrast: Preferred > neutral: MDD + MJ > Other Groups: R middle and inferior frontal gyrus R postcentral gyrus L precentral and postcentral gyrus L cingulate gyrus R inferior frontal and precentral gyrus extending to claustrum and putamen MDD + MJ < Other Groups: None
Gilman et al., 2016a [54]	N = 20 social CU young adults and N = 20 non-using CON young adults ages 18–25-yrs.	CU: 50% CON: 50%	CU: 20.6 yrs. CON: 21.5 yrs.	All members of CU group reported weekly CU	Community	≥12-h	3.0 T	Social-influence Decision-Making task (using graph to represent peer choices)	Primary contrast: Social influence vs. No-influence contrast (during choice phase); Secondary contrast(s): Congruent vs. incongruent choice contrast (during choices w/social influence stimuli) and Win vs. Loss feedback contrast	Analysis: Wb, NAc ROI, two-way ANOVAs FWHM: 5 mm MC: k > 20 voxels, voxelwise *p* < 0.005, Z > 2.6, cluster corrected *p* < 0.05 Sampling: CU vs. TD group comparison	Primary contrast (social influence): MJ > CON: L frontal pole L superior temporal gyrus L superior parietal gyrus MJ < CON: none Secondary contrasts: Incongruent vs. congruent choice: No group differences Win vs. Loss feedback: No group differences
Gilman et al., 2016b [55]	N = 20 social CU young adults and N = 23 non-using CON young adults ages 18–25-yrs.	CU: 45% CON: 48%	CU: 20.6 yrs. CON: 21.6 yrs.	All members of CU group reported weekly CU	Community	≥12-h	3.0 T	Social-influence decision-making task (using peer images as social stimuli)	Social influence vs. No-influence contrast (during Choice phase); Congruent vs. incongruent choices (during choices w/social influence stimuli)	Analysis: Wb, NAc ROI FWHM: 5 mm MC:, Z > 2.3, cluster corrected *p* < 0.05 Sampling: CU vs. TD group comparison	Social influence vs. No-influence contrast: MJ > CON: R Caudate MJ < CON: None Incongruent vs. congruent choices: MJ > CON: None MJ < CON: None
Gilman et al., 2016c [56]	N = 20 heavy CU and N = 22 non-using CON young adults ages 18–25-yrs.	CU: 45% CON: 50%	CU: 21.4 yrs. CON: 20.4 yrs.	All members of CU group reported weekly use; 50% of CU group met criteria for current CUD	Community	≥12-h	3.0 T	Cyberball task (social exclusion paradigm)	Primary contrast: exclusion vs. inclusion contrast Secondary contrast(s): Inclusion vs. exclusion (social inclusion) and reinclusion vs. inclusion (response to reinclusion following exclusion)	Analysis: WB, right insula and ACC ROIs, Mixed effect analysis FWHM: 5 mm MC: Z > 2.3, cluster corrected *p* < 0.05 Sampling: CU vs. TD group comparison	Primary contrast: Exclusion vs. fair play: MJ > CON: none MJ < CON: R insula R orbitofrontal cortext/insula Secondary contrasts: No MJ vs. CON group differences for secondary contrasts (social inclusion or win vs. loss feedback)
Hatchard et al., 2014 [57]	N = 10 regular CU and N = 14 healthy CON young adults ages 19–21 yrs.	CU: 60% CON: 64%	CU: 20.0 yrs. CON: 20.0 yrs.	All CU participants were regular users defined as smoking > 1 joint per week for at least 3-yrs.	Community (Mixed risk community sample from Ottowa Prenatal Prospective Study)	Ad-lib use	1.5 T	Counting Stroop Interference task	Incongruent (‘Numbers’)—Congruent (‘Animals’) contrast across all trials	Analysis: WB, independent sample *t*-tests FWHM: 8 mm MC: WB voxelwise *p* < 0.001, cluster corrected pFWE < 0.05 via SPM Sampling: CU vs. TD group comparison	Incongruent—Congruent: contrast: MJ > CON: R rolandic operculum R cerebellar tonsil R postcentral gyrus Cingulate gyrus L postcentral gyrus R SMA MJ < CON: None
Heitzeg et al., 2015 [58]	N = 20 heavy CU young adults and N = 20 healthy CON young adults ages 17–22 yrs.	CU: 60% CON: 70%	CU: 19.8 yrs. CON: 20.5 yrs.	All CU participants had > 100 lifetime use episodes	Community (Mixed risk community sample from Michigan Longitudinal Study)	≥48-h	3.0 T	Emotion arousal word task	Negative vs. neutral and positive vs. neutral word contrasts	Analysis: WB + amygdala ROI, GLM FWHM: 6 mm MC: WB: *p* < 0.005; k > 77 voxels (est. using AlphaSim) Sampling: CU vs. TD group comparison	Negative vs. neutral contrast: MJ > CON: none MJ < CON: R caudal dlPFC R MTG/STG R cuneus/lingual gyrus R STG/insula R amygdala L amygdala Positive vs. neutral contrast: MJ > CON: R dlPFC MJ < CON: R IPL R Amygdala L Amygdala
Jacobsen et al., 2007 [36]	N = 20 MJ + TOB users and N = 25 TOB users with limited MJ history scanned at satiety and 24-hr abstinence from nicotine.	MJ + TOB: 25% TOB: 28%	MJ + TOB: 17.3 yrs. TOB: 17.0 yrs.	MJ + TOB group had ≥60 MJ use episode; Both MJ + TOB and TOB groups were daily cigarette smokers	Community	≥30-day	1.5 T	Auditory N-Back task	Task Contrast: WM-load (2-back vs. 1-back); Within-Subject Tobacco smoking status contrast (Ad-lib tobacco smoking vs. 24-hr tobacco abstinence)	Analysis: WB, Linear Mixed Regression Models FWHM: 3.125 MC: WB voxelwise *p* < 0.001, k > 8 voxels Sampling: CU vs. TD group comparison	Group-by-WM-load (2-back vs. 1-back): No MJ + TOB vs. TOB-only group differences Group-by-WM load-by-smoking condition interaction effect in L IPL/STG, R STG, R posterior insula, L posterior cingulate. Group findings based upon the 3 contrasts showed: 2-Back vs. 1-Back: MJ + TOB _24-hr-Abst_ > TOB _24-hr-Abst_ L IPL/STG R posterior insula R STG L posterior cingulate Other MJ + TOB vs. TOB state-by-trait comparisons showed no differences
Jager et al., 2010 [59]	N = 21 regular CU male youth and N = 24 healthy CON male youth	CU: 100% CON: 100%	CU: 17.2 yrs. CON: 16.8 yrs.	CU participants had at least 200 lifetime CU episodes	Community (two sites: Netherlands and United States)	≥24-h	3.0 T and 3.0 T	Sternberg Verbal WM task and Pictorial Associative Memory Task	WM vs. Control; Practiced WM vs. Control; Novel WM vs. Control; Associative learning (collapsed across AL and AR conditions) vs. Classification phase;	Analysis: WB + ROI FWHM: 8 mm MC: WB, pFWE < 0.05 Sampling: CU vs. TD group comparison	WM (collapsed across Practiced and Novel WM trials) vs. Control condition contrast: No group differences PAMT: No group differences
Jager et al., 2013 [27]	N = 23 regular CU male youth and N = 24 healthy CON male youth	CU: 100% CON: 100%	CU: 17.2 yrs. CON: 16.8 yrs.	CU participants had at least 200 lifetime CU episodes	Community (two sites: Netherlands and United States)	≥24-h	3.0 T and 3.0 T	Monetary Incentive Delay (MID) task	Anticipation phase contrast: Reward vs. neutral anticipation Feedback phase: win vs. loss feedback during reward trials	Analysis: Analysis: WB + caudate, putamen, VS ROIs; GLM repeated measure analyses FWHM: 8 mm MC: WB: pFWE < 0.05 Sampling: CU vs. TD group comparison	Whole-brain voxel-wise analyses showed no MJ vs. CON group differences in reward vs. neutral anticipation contrast or win vs. loss feedback contrast.
Kroon et al., 2021 [60]	N = 36 daily CU youth and N = 33 healthy CON youth	CU: 53% CON: 49%	CU: 21.0 yrs. CON: 21.0 yrs.	All CU youth reported daily or near daily use. Mean CUDIT score of CU participants was 13.0.		≥24-h	3.0 T	N-back flanker WM task with neutral and cannabis flankers	3 contrasts of interest: cannabis (c) > neutral (n) flanker contrast (main effect of flanker); 2-back (2) > 1-back (1) contrast (i.e., main effect of WM); and flanker-by-WM-interaction contrast ((2c > 1c) > (2n > 1n))	Analysis: WB, Mixed effect group analysis and independent sample *t*-tests FWHM: 5 mm MC: WB k >10 voxels, z > 2.3, cluster corrected *p* < 0.05 Sampling: CU vs. TD group comparison	Flanker effect (c > n): MJ > CON: none MJ < CON: none WM performance: No MJ vs. CON group differences in accuracy or reaction time on N-back task. WM effect (2 vs. 1): MJ > CON: none MJ < CON: L STG L MTG L angular gyrus Flanker-by-WM effect: MJ > CON: none MJ < CON: L thalamus L operculum L insula R SPL R SMG R PCG
Leiker et al., 2019 [61]	N = 104 youth ages 14–18-years with variable levels of CUD and AUD severity recruited from residential program and community	Total Sample: 64%	Total Sample: 16.0 yrs.	Mean CUDIT score of Total Sample = 5.5	Community/Clinical	≥30-days	3.0 T	Emotional faces task	Fearful vs. happy vs. neutral faces contrasts	Analysis: WB, ANCOVAs FWHM: 6 mm MC: WB k > 24 voxels, voxelwise: *p* < 0.001, cluster corrected *p* < 0.05 (via AFNI3dClustSIM) Sampling: Combined Sample of CU and TD youth	Whole-brain meta-regression: Emotional vs. neutral faces contrast: Main effect of CUD symptoms: Negative association between CUDIT scores and BOLD response to emotional face stimuli in L rostromedial PFC including left caudal, ACC regions. Traditional group-based analyses: CUD < CON: R rostrommedial PFC/ACC
Lopez-Larson et al., 2012 [62]	N = 24 regular CU and N = 24 healthy CON youth ages 16–22 yrs.	CU: 92% CON: 71%	CU: 18.2 yrs. CON: 18.0 yrs.	All CU participants had at least 100 lifetime CU episodes	Community	≥12-h	3.0 T	Standard bilateral finger tapping task	No contrast, BOLD response during finger tapping compared between groups	Analysis: WB + ROI FWHM: 8 mm MC: WB: k > 20 voxels, Cluster corrected *p* < 0.005 Sampling: CU vs. TD group comparison	Finger tapping related BOLD activation: MJ > CON: R middle occipital lobe MJ < CON: R cingulate gyrus
May et al., 2020 [63]	N = 13 CAN + ALC-SUD, N = 16 CAN + ALC-EXP, and N = 18 CON adolescents ages 15–17 years	CAN + ALC-SUD: 69% CAN + ALC-EXP: 75% CON: 72%	CAN + ALC-SUD: 16.6 yrs. CAN + ALC-EXP: 16.7 yrs. CON: 16.3 yrs.	CAN + ALC-SUD: 92.3% CUD diagnoses and 61.5% AUD diagnoses with Mean lifetime CU episodes: 467.9 CAN + ALC-EXP: Mean lifetime CU episodes: 39.4 CON: Mean lifetime CU episodes: 0.1	Community (California high schools)	≥72-h	3.0 T	Drug Cue Breathing fMRI paradigm that paired a cannabis/alcohol drug cue reactivity task with anticipation and experience of aversive interoceptive stimulus (inspiratory breathing load)	9 task conditions: anticipation neutral images, anticipation substance images, anticipation scrambled images, breathing load neutral images, breathing load substance images, breathing load scrambled images, neutral images only, substance images only, and scrambled images only Contrasts of interest: BOLD signal differences between 3 groups (CAN + ALC-SUD vs. CAN + ALC-EXP vs. CON participants) across breathing load and cue image type conditions; group-by-image type (substance vs. neutral contrast); group-by-interoceptive condition (no breathing load [anticipation] vs. breathing load); and Group-by-Image type-by Interoceptive condition	Analysis: WB, linear mixed effect models FWHM: 6 mm MC: k > 1280 µL (20 voxels), voxelwise *p* < 0.002, cluster corrected *p* < 0.05 (via AFNI3dClustSIM) Sampling: CU vs. TD group comparison	Main Group and Image type interaction effects: No group differences were observed in main group comparison or in group-by-image type or group-by-image type-by interoceptive condition interactions. Group-by-interoceptive condition effect (anticipation vs. breathing load): R amgydala L IFG R posterior cingulate L parahippocampal gyrus Post Hoc Pairwise Group Comparisons from Group-by-Interoceptive Condition effect: Anticipation condition: CAN + ALC-SUD > CAN +ALC-EXP: R amgydala CAN + ALC-EXP < CON: L parahippocampal gyrus CAN + ALC-SUD < CAN +ALC-EXP: None Breathing load condition: CAN + ALC-SUD > CAN +ALC-EXP None CAN + ALC-SUD < CAN +ALC-EXP R amgydala CAN + ALC-SUD < CAN +ALC-EXP and CON: L inferior frontal gyrus L parahippocampal gyrus
Migliorini et al., 2013 [64]	N = 15 adolescents with SUD and N = 17 healthy CON adolescents ages 15–17 yrs.	SUD: 67% CON: 65%	SUD: 16.5 yrs. CON: 16.8 yrs.	All SUD participants met criteria for current CUD and/or AUD; 73% of SUD participants met criteria for CUD	Community (California high schools)	≥72-h	3.0 T	Interoceptive Stimulation task (Soft Touch task)	Soft touch vs. anticipation contrast	Analysis: WB + striatal and ant/post. Insula ROIs, LME FWHM: 4 mm MC: WB: k > 512μL (8 voxels), cluster corrected *p* < 0.05 based on MCS Sampling: CU vs. TD group comparison	Group main effect: SUD > CON: none SUD < CON: L posterior insula L cuneus R inferior temporal gyrus Group by condition interaction: SUD > CON: None SUD < CON: L postcentral gyrus R precentral gyrus L posterior insula L precentral gyrus R middle frontal gyrus L postcentral gyrus R medial frontal gyrus L cerebellar lingual gyrus R cingulate gyrus R cuneus R medial frontal gyrus L precuneus
Padula et al., 2007 [29]	N = 17 CU adolescents and N = 17 healthy CON adolescents ages 16–18 yrs.	CU: 82% CON: 71%	CU: 18.1 yrs. CON: 17.9 yrs.	CU participants had an average of 477 lifetime CU episodes	Community	>28-days	1.5 T	Spatial WM N-Back task	SWM vs. vigilance condition (1-Back vs. 0-Back)	Analysis: WB, Regressions FWHM: 5 mm MC: WB; k > 50 voxels; Cluster-wise *p* < 0.05 corrected using MCS Sampling: CU vs. TD group comparison	1-Back vs. 0-Back: MJ > CON: R claustrum, putamen, caudate, thalamus, globus pallidus, insula, globus pallidus R precuneus, superior parietal lobule, postcentral gyrus L superior parietal lobule, precuneus MJ < CON: none
Raymond et al., 2020 [65]	N = 17 regular CU young adults and N = 14 non-using CON young adults	CU: 47% CON: 43%	CU: 21.2 yrs. CON: 22.5 yrs.	The CU group used an avg. of 5.2 days per week and had mean CUDIT score of 13.4	Community	≥12-h	3.0 T	Balloon Analogue Risk Task (BART)	Primary contrast: Risk taking condition which reflects the choice to inflate the balloon x the probability of explosion ChooseInflate*P(explode)	Analysis: WB, GLM FWHM: 8 mm MC: WB k > 150 voxels, *p* < 0.001 Sampling: CU vs. TD group comparison	Risk condition: MJ > CON: R lateral posterior PFC extending into frontal eye fields
Schweinsburg et al., 2005 [18]	N = 15 AUD, N = 15 comorbid CUD + AUD, and N = 19 healthy CON adolescents ages 15–17 yrs.	AUD: 67% CUD + AUD: 67% CON: 58% 15(66.7%)	AUD: 16.8 yrs. CUD+ AUD: 16.9 yrs. CON: 16.5 yrs.	CUD+ AUD participants met criteria for current CUD and AUD and had >100 lifetime CU episodes	Community	≥48-h	1.5 T	Spatial WM task	WM vs. Simple Attention/Vigilance Trial contrast	Analysis: WB, one-way ANOVAs FWHM: 3.5 mm MC: WB: k > 1072 μL (25 voxels), cluster corrected *p* < 0.016 Sampling: CU vs. TD group comparison	SWM > Attention: MJ + AUD > CON: R superior frontal and middle frontal gyri MJ + AUD < CON: R inferior frontal gyrus R superior temporal and supramarginal gyrus SWM < Attention: MJ + AUD > CON: L inferior frontal gyrus B/L inferior frontal and anterior cingulate gyri MJ + AUD < CON: None
Schweinsburg et al., 2008 [17]	N = 15 heavy CU youth and N = 17 healthy CON youth ages 16–18 yrs.	CU: 73% CON: 71%	CU: 18.1 yrs. CON: 17.9 yrs.	CU participants had an average of 480.7 lifetime use episode	Community	≥28-days	1.5 T	Spatial WM task	SWM vs. Attention/Vigilance trial contrast	Analysis: WB, independent sample *t*-tests FWHM: 5 mm MC: WB: k > 1328 μL, cluster corrected *p* < 0.05 Sampling: CU vs. TD group comparison	SWM > Vigilance: MJ > CON: R superior parietal gyrus MJ < CON: R middle frontal gyrus SWM < Vigilance: MJ > CON: R cuneus L lingual gyrus and cuneus MJ < CON: none
Schweinsburg et al., 2010 [66]	N = 13 recent CU youth, N = 13 abstinent CU youth, and N = 18 healthy CON youth ages 15–18 yrs.	Recent CU: 69% Abstinent CU: 69% CON: 61%	Recent CU: 17.1 yrs. Abstinent CU: 17.6 yrs. CON: 17.3 yrs.	Recent CU: 342 lifetime CU episodes Abstinent CU: 515 lifetime CU episodes	Community	Recent Users: ≥24-h; Abstinent Users: ≥27-days	1.5 T	Spatial WM task	SWM vs. vigilance contrasts	Analysis: WB, independent sample *t*-tests FWHM: 5 mm MC: k > 1328 µL (49 voxels), t > 2.06, cluster corrected *p* < 0.05 Sampling: CU vs. TD group comparison	Recent CU > Abstinent CU: Medial cingulate MFG L SFG and MFG Bilateral mPFC Bilateral insula L precentral gyrus R IFG Abstinent CU > Recent CU: Right precentral gyrus Post Hoc Pairwise Comparisons: Recent CU showed increased bilateral mPFC and insula activation to SWM compared to vigilance condition, while Abstinent CU showed decreased activation and CON showed no activation differences in these regions to SWM vs. vigilance condition. Recent CU showed decreased R precentral gyrus activation to SWM compared to vigilance condition, while abstinent CU and CON showed no activation difference during SWM vs. vigilance in this region.
Schweinsburg et al., 2011 [38]	N = 8 MJ users, N = 16 BD, N = 28 MJ + BD, and N = 22 healthy CON adolescents	MJ: 50% BD: 81% MJ+ BD: 82% CON: 73%	MJ: 18.1 yrs. BD: 18.1 yrs. MJ + BD: 18.0 yrs. CON: 17.6 yrs.	MJ and MJ + BD groups both had >180 lifetime MJ use episodes	Community	≥21-days	3.0 T	Verbal Paired Associations Test	Primary contrast: BOLD response to novel word pairs	Analysis: WB + hippocampal ROI, ANOVAs FWHM: 5 mm MC: WB k > 1512 µL, cluster corrected *p* < 0.05 Sampling: CU vs. TD group comparison	Main effect of Marijuana Use: (Examined by collapsing across Subgroups): MJ and BD + MJ > CON and BD: None MJ and BD + MJ < CON and BD: None Drinking x Marijuana Interaction: (Whole-brain): L superior and middle frontal gyri R inferior and middle frontal gyri R superior and middle frontal gyri Medial cuneus/lingual gyrus Post Hoc Pairwise Group Comparisons from Drinking X MJ interaction: MJ > CON and BD + MJ: L superior and middle frontal gyri MJ and BD > CON: R superior and middle frontal gyri MJ > CON: R middle and inferior frontal gyri MJ and BD < CON: b/l cuneus and lingual gyri
Smith et al., 2010 [30]	N = 10 current CU young adults and N = 14 non-using CON young adults ages 19–21 yrs.	CU: 60% CON: 64%	CU: 20.0 yrs. CON: 20.0 yrs.	All CU participants used cannabis weekly	Community	Ad-lib use	1.5 T	N-Back WM task (Visuospatial 2-back task)	2-back vs. 0-back contrast	Analysis: WB, two-sample *t*-tests FWHM: 8 mm MC: WB: Cluster corrected *p* < 0.05 Sampling: CU vs. TD group comparison	Behavioral: No group differences in WM performance on 2-back and 0-back 2-back vs. 0-back fMRI contrast: MJ > CON: R inferior frontal gyrus R superior temporal gyrus and temporal pole R cingulate gyrus MJ < CON: None
Tapert et al., 2007 [19]	N = 16 CU adolescdents and N = 17 healthy CON adolescents ages 16–18 yrs.	CU: 75% CON: 71%	CU: 18.1 yrs. CON: 17.9 yrs.	CU group endoresed and avg. of 500 lifetime CU episodes	Community (California high schools)	>28-days	1.5 T	Go/No-Go task	Inhibition (No-Go) Trials vs. Baseline contrast (primary outcome); Go Trials vs. Baseline contrast	Analysis: WB, independent sample *t*-tests FWHM: 3.5 mm MC: WB: k > 22 voxels, Cluster corrected *p* < 0.05 Sampling: CU vs. TD group comparison	Inhibition (No-Go) trial contrast: MJ > CON: R Superior and middle frontal gyrus R middle frontal gyrus and insula L middle and superior frontal gyri b/l medial frontal cortex R inferior and superior parietal lobes L inferior and superior parietal lobes R lingual and middle occipital gyrus MJ < CON:None Go trial contrast; MJ > CON: R inferior frontal gyrus and insula R superior and middle frontal gyurs R superior parietal lobe R inferior parietal lobe R medial precuneus MJ < CON: None
Tervo-Clemmens et al., 2018 [67]	N = 85 participants completed a baseline MRI session at age 12-yrs and then followed up at age 15-yrs. At follow-up: N = 22 participants were CU and N = 63 were non-CU	CU group: 55% Non-CU group: 46%	CU group: 15.6 yrs. Non-CU group: 15.6 yrs.		Community (longitudinal sample of US youth enriched for SUD risk)	≥24-h	3.0 T	Spatial WM task	BOLD response during successful/correct WM trials at age 12 baseline visit and age 15 follow-up visit for MJ preusers/users and non-using CON.	Analysis: WB, Multivariate Models FWHM: 5 mm MC: WB: k > 11, voxelwise *p* < 0.005, cluster corrected pFDR < 0.05 (done w/AFNI 3dClustSIM) Sampling: CU vs. TD group comparison	Baseline group comparison: MJ Pre-Users > CON Non-Pre-Users: b/l MFG L inferior parietal lobule Paracentral lobule/cingulate gyrus MJ Pre-Users < CON Non-Pre-Users: b/l lingual gyrus L precuneus Pre-SMA L lateral occipital gyrus Follow-up group comparison: MJ users > CON: None MJ users < CON: R Cuneus Post hoc analysis of follow-up data showed a significant negative correlation between BOLD response in the R cuneus cluster and cannabis dose Group-by-Time effect: Posterior cingulate cortex
Thayer et al., 2015 [68]	N = 80 high-risk adolescents with variable cannabis and alcohol use behaviors	Total sample: 74%	Total sample: 15.9 yrs.	Total sample reported a past-3-month avg. of 7–9 hits of MJ on 4–5 occasions per month and 2–3 drinks per month	Justice system (juvenile justice program in SW United States)	NP	3.0 T	Stroop Color-Word Interference task	Contrasts of interest: Incongruent—Neutral and Incongruent—Congruent contrasts during correct trials	Analysis: WB, GLM FWHM: 8 mm MC: WB: k > 2496 µL, voxelwise: *p* < 0.005, Cluster corrected *p* < 0.05 (done w/AFNI 3dClustSIM) Sampling: Combined Sample	Incongruent—Neutral contrast: No main or interaction effects of MJ frequency on BOLD response Incongruent—Congruent contrast: No main or interaction effects of MJ frequency on BOLD response
Zhou et al., 2019 [69]	N = 26 Dependent MJ users, N = 25 Non-Dependent MJ users, and N = 52 healthy CON youth	Dep MJ: 100% Non-Dep MJ: 100% CON: 100%	Dep MJ: 22.9 yrs. Non-Dep MJ: 21.5 yrs. CON: 23.2 yrs.	Dep. MJ: 1538 g lifetime use Non-Dep MJ: 985 g lifetime use	Community	≥24-h	3.0 T	Drug Cue-reactivity task	Cannabis cue vs. neutral cue contrast	Analysis: WB + dorsal and ventral striatal ROI, mixed ANOVAs, *t*-tests FWHM: 6 mm MC: voxelwise *p* < 0.001, Cluster corrected pFWE < 0.05 Sampling: CU vs. TD group comparison	Cannabis vs. Neutral cue: Non-dependent MJ > CON: Ventral caudate Nucleus accumbens Superior parietal lobe and precuneus Non-dependent MJ < CON: None Dependent MJ > CON: Limbic lobe extending to temporal, occipital, and parietal lobes R inferior frontal gyrus extending to middle frontal gyrus L superior frontal gyrus extending to middle frontal gyrus L IPL extending to posterior cingulate cortex and precuneus L fusiform R inferior frontal gyrus Medial PFC extending to anterior cingulate cortex L inferior frontal gyrus extending to middle frontal gyrus L inferior frontal gyrus Dependent MJ < CON: None
Zimmerman et al., 2017 [70]	N = 23 regular recreational CU young adults and N = 22 non-using matched CON young adults	CU: 100% CON: 100%	CU: 21.2 yrs. CON: 21.1 yrs.	All CU participants used cannabis >3 times per week over the past-yr. and had >200 lifetime use episodes	Community	≥48-h	3.0 T	Cognitive reappraisal task	Primary contrast: distance vs. baseline contrast (emotion regulation using cognitive reappraisal). Secondary contrast: spontaneous negative vs. baseline contrast (emotional reactivity to negative stimuli)	Analysis: WB + amygdala ROI, GLM, also seed-based amygdala-ROI FC analysis FWHM: 8 mm MC: WB: pFWE < 0.05 Sampling: CU vs. TD group comparison	Distance vs. bsl contrast: MJ > CON: b/l precentral gyrus R superior frontal gyrus L mid-cingulate/SMA L precentral gyrus R amygdala MJ < CON: None Emotion reactivity: Spontaneous negative vs. bsl: No group differences

Note: Numbers in ( ) following each study represent the citation number linked to the Reference section. Abbreviations: ACC = anterior cingulate cortex, ALC-alcohol, ANOVA = analysis of variance, ANCOVA = analysis of covariance, AUD = alcohol use disorder, AUDIT = alcohol use disorder identification test, BD = binge drinking, BOLD = blood-oxygen-level-dependent, BSL = baseline, CON = Control, CU = Cannabis Using, CUD = cannabis use disorder, CUDIT = cannabis use disorder identification test, Dep. = Dependent Cannabis users (i.e., cannabis users who meet DSM-IV criteria for Cannabis Dependence), DM = decision making, EXP—drug experimenters, FC = functional connectivity, FEW = family-wise error correction, FWHM = full width half maximum (spatial smoothing of images), GLM = general linear model, IFG = inferior frontal gyrus, IPL = inferior parietal lobule, L = left, MC = multiple comparisons (approach each study takes to threshold and control for multiple comparisons), MCS = monte carlo simulations, MDD = major depressive disorder, MJ = marijuana or cannabis using participant group, MTG = middle temporal gyrus, NAC = nucleus accumbens, Non-dep. = Non-Dependent Cannabis users (i.e., cannabis users who do not meet DSM-IV criteria for cannabis dependence), oFC = orbitofrontal cortex, PCC = posterior cingulate cortex, PFC = prefrontal cortex, R = right, ROI = region of interest analysis, RPE = reward prediction erro, SMG = supramarginal gyrus, SEM = structural equation modeling, SMA = supplemental motor areal, SPL = superior parietal lobule, STG = superior temporal gyrus, SWM = spatial working memory, TD = typically developing youth, TOB = tobacco smoking/cigarette smoking, WB = whole-brain analysis, WM = working memory, VTA = ventral tegmental area, yrs. = years.

**Table 2 brainsci-12-01281-t002:** Meta-analysis of fMRI studies comparing CU and TD youth across all studies and by cognitive domain.

			MNI Coordinates						
Cluster #, Label	BA	Voxels	x	y	z	SDM-Z	*p*-Value		
**All studies (45 studies)**									
CU > TD youth									
None									
CU < TD youth									
None									
**Executive Function/Cognitive Control studies (16 studies)**									
CU > TD youth									
Cluster #1 Right rostral mPFC	10, 11	5	4	60	−4	2.615	0.0044618		
CU < TD youth									
None									
**Social Cognition/Emotion Processing studies (9 studies)**									
CU > TD youth									
None									
CU < TD youth									
Cluster #2 Left dorsal mPFC Right dorsal mPFC Left dorsal ACC Right dorsal ACC	10, 32	64	2	50	22	−3.100	0.0009676		
**Reward Processing studies (8 studies)**									
CU > TD youth									
None									
CU < TD youth									
None									
**Drug Cue Reactivity studies (5 studies)**									
CU > TD youth									
None									
CU < TD youth									
None									

Note: SDM meta-analyses were carried out in SDM-PSI.v.6.21 on fMRI studies comparing CU and TD youth reporting. Talaraich or MNI coordinates with threshold set at *p*-value < 0.005. Coordinates shown are MNI. Abbreviations: ACC = Anterior Cingulate Cortex; mPFC = medial prefrontal cortex; CU = Cannabis Using; TD = Typically Developing; BA = Broadman’s area.

## Data Availability

All data elements used in this meta-analytic study are publicly available except for the individual study data and contrast maps obtained following personal communication with study authors. Data elements for the meta-regression analyses can be found in the online Appendix A.

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
