# Peer review of "A Meta-Analysis of fMRI Studies of Youth Cannabis Use: Alterations in Executive Control, Social Cognition/Emotion Processing, and Reward Processing in Cannabis Using Youth"

_brainsci, 2022, doi:10.3390/brainsci12101281_

Round 1

Reviewer 1 Report

Thank you for the opportunity to revise this manuscript and congratulations to the authors for the relevant work.

The paper comprehensively examines convergent fMRI findings from tasks indexing regional brain activations in various cognitive domains, in youth with cannabis use, through metanalysis. As the authors underline in the Introduction, this is a topic that to date has not been clearly investigated. At the same time, this is a very timely topic, given the diffusion of cannabis use and its effects in terms of health.

Methodology of the study is adequate, clearly and well described. The authors properly used the PRISMA guidelines and method.

Results are well presented, also using effective images.

Discussion is complete, also providing limitation section.

The only consideration, regarding the Keywords: according to the Journal instructions, keywords can be up to 10. The authors included 12

Author Response

Dear Reviewer 1:  Thank you for your thoughtful review and feedback.  In response to your comments, we have removed 2 keywords that were duplicative.  Our revised manuscript now has the correct number of keywords (10 or fewer).  

Reviewer 2 Report

The study qualitatively and quantitatively summarized the functional neuroimaging studies that examine neural correlates of cannabis use in adolescents and young adults using signed differential mapping meta-analytic approach.

Suggestions: A risk of bias analysis for the study is appreciable, as is a GRADE-based level of evidence analysis.

The study is very well conducted, and it was gratifying to review it, I can indicate acceptance for manuscript.

Final considerations

The manuscript is interesting paper, in this form satisfy the scientific standards and quality  of Journal.

Sincerely.

Author Response

  1.  

eFig S2. Caption: Funnel Plots for Primary Meta-analyses of Executive Function/Cognitive Control Domain (A) and Social Cognition/Emotion Processing Domain (B). Funnel plots were created using SDM software and plotted the effect estimate (standardized BOLD signal difference between CU and TD participants) on the X-axis and the variance on the Y-axis for each study included in the primary meta-analyses focused on EF/CC and SC/EM domains.  Results of these funnel plots show symmetric distribution of studies suggesting low evidence for bias in our two primary meta-analytic results.  Using SDM’s Metabias calculation tool, the Risk of Bias for the EF/CC domain meta-analysis is: 0.86, z: 1.03, df: 14, p=0.301 and the Risk of Bias for the SC/EM domain meta-analysis is: -0.19, z: -0.20, df: 7, p=0.838.

Point 2:  Reviewer 2 suggested that the project team consider reporting a GRADE-based level of evidence analysis for the study.

Response 2:  Your suggestion to consider including a level of evidence analysis for the meta-analytic report is a thoughtful one.  Problematically, the GRADE-based strength of evidence approach was designed to assess the quality of evidence from clinical intervention studies does not adapt well to observational neuroimaging studies with BOLD signal outcomes.  The field of neuroimaging has established guidelines/standards for reporting results of fMRI outcomes (see OHBM Committee on Best Practice in Data Analysis and Sharing [COBIDAS] Report publised in 2016: https://www.humanbrainmapping.org/i4a/pages/index.cfm?pageid=3728) but there remain multiple approaches/methods for how data are collected, preprocessed, and analyzed that likely impact findings, and much work is needed to determine best practices and landmarks for strength of evidence in this space.  Still, you bring up a good point with this comment.  Related to your point, in the supplement (pp. 6) we include a section describing experimental design, analytic approaches, sample characteristics and discussing the heterogeneity across included studies. 
